# CABra: a novel large-sample dataset for Brazilian catchments

André Almagro[1], Paulo Tarso S. Oliveira[1], Antônio Alves Meira Neto[2], Tirthankar Roy[3] & Peter Troch[4]

[1]Faculty of Engineering and Geography, Federal University of Mato Grosso do Sul, Campo Grande, MS, Brazil.
[2]Institute of Climate Studies, Federal University of Espírito Santo, Vitória, ES, Brazil.
[3]Civil and Environmental Engineering, University of Nebraska-Lincoln, Omaha, NE, United States.
[4]Department of Hydrology and Atmospheric Sciences, The University of Arizona, Tucson, AZ, United States.

*Correspondence to*: André Almagro (andre.almagro@gmail.com)

**Abstract.** In this paper, we present the Catchments Attributes for Brazil (CABra), which is a large-sample dataset for Brazilian catchments that includes long-term data (30 years) for 735 catchments in eight main catchment attribute classes
(climate, streamflow, groundwater, geology, soil, topography, land-cover, and hydrologic disturbance). We have collected and synthesized data from multiple sources (ground stations, remote sensing, and gridded datasets). To prepare the dataset, we delineated all the catchments using the Multi-Error-Removed Improved-Terrain Digital Elevation Model and the coordinates of the streamflow stations provided by the Brazilian Water Agency, where only the stations with 30 years (1980-2010) of data and less than 10% of missing records were included. Catchment areas range from 9 to 4,800,000 km² and the
mean daily streamflow varies from 0.02 to 9 mm day$^{-1}$. Several signatures and indices were calculated based on the climate and streamflow data. Additionally, our dataset includes boundary shapefiles, geographic coordinates, and drainage area for each catchment, aside from more than 100 attributes within the attribute classes. The collection and processing methods are discussed along with the limitations for each of our multiple data sources. The CABra intends to improve the hydrology-related data collection in Brazil and pave the way for a better understanding of different hydrologic drivers related to climate,
landscape, and hydrology, which is particularly important in Brazil, having continental-scale river basins and widely heterogeneous landscape characteristics. In addition to benefitting catchment hydrology investigations, CABra will expand the exploration of novel hydrologic hypotheses and thereby advance our understanding of Brazilian catchments' behavior. The dataset is freely available at https://doi.org/10.5281/zenodo.4070146 and https://thecabradataset.shinyapps.io/CABra/.

## 1 Introduction

The integrated assessment of large-sample catchment attributes is fundamental for the description and classification of landscape properties, leading to an improved understanding of similarities (or dissimilarities) between catchments. Large-sample catchment hydrology is essential in terms of hydrological processes understanding (Addor et al., 2020; Beven et al., 2020). It provides an attractive venue for general inferences that would otherwise be impossible to study based on individual or small groups of catchments, aside from allowing the testing of new and existing hypotheses in hydrologic sciences (Addor
et al., 2017; Gupta et al., 2014; Lyon and Troch, 2010; Wagener et al., 2007).

A classic example of a large catchment-scale dataset is the Model Parameter Estimation Experiment (MOPEX) (Duan et al., 2006; Schaake et al., 2006), with hydrologic time series from 438 catchments located within the continental US (CONUS). The MOPEX dataset has been used in several studies supporting theoretic and modeling advances in hydrologic sciences (Ao et al., 2006; Ren et al., 2016; Sawicz et al., 2011). A more recent example is the Catchment Attributes and MEteorological for Large-sample Studies (CAMELS, Addor et al. (2017)) consisting of a set of daily hydrometeorological time series data for 671 small- to medium-sized catchments for the CONUS, aside from several landscape and climate related attributes. The CAMELS initiative has been widely used and other large-sample datasets have been recently developed following the CAMELS format, such as CAMELS-GB for Great Britain, covering 671 catchments, CAMELS-CL for Chile, covering 516 catchments, and CAMELS-BR for Brazil, covering 897 catchments. A list of available large-sample datasets can be found in Addor et al. (2020).

Brazil is a country with continental dimensions, hosting a wide range of climates, soils, geology, and land-cover types. Despite covering almost 50% of South America and hosting between 12% and 18% of the world's renewable freshwater (Rodrigues et al., 2015; UNEP and ANA, 2007), Brazil suffers from scarce allocation of funds for hydrological monitoring services, which creates great challenges for the proper monitoring of the quality and quantity of its water resources. While the density of streamflow gauges falls below the standards recommended by the World Meteorological Organization (WMO) of 1 station for each 1,000 km², hydrologic observations are often discontinued and lack proper length (ANA, 2019a; WMO, 2010). An integrated dataset containing multiple levels of environmental information can be of extreme importance to leverage investigations in hydrology and related disciplines within the Brazilian territory.

Recently, two large-sample datasets for catchment attributes were developed for Brazil: the Catchment Attributes for Brazil (CABra) (first introduced in Oliveira et al., 2020) and the Catchment Attributes and MEteorology for Large-sample Studies (CAMELS-BR) (Chagas et al., 2020). Even though both datasets aim to fill the lack of hydrological data access in Brazil, the data sources, quality control, number, and types of attributes differ significantly. To address the similarities and differences between both datasets, an extensive discussion comparing CAMELS-BR and CABra is also presented in our study.

In this paper, we present the CABra dataset, which is a comprehensive, large-sample dataset for catchment attributes in Brazil. We have synthesized several multi-source data from eight main attribute classes (topography, climate, streamflow, groundwater, soil, geology, land-use and land-cover, and hydrologic disturbance) for 735 catchments in Brazil. Our dataset covers all Brazilian administrative and hydrographic regions as well as its biomes. We have delimited all the catchments using an error-corrected digital elevation model employing automatic drainage area delineation methods. For the area-averaged attributes, we have used national datasets from the Brazilian Water Agency (ANA), Brazilian Agricultural Research Corporation (EMBRAPA), and Xavier et al. (2016), and widely used global datasets, such as ERA5, SoilGrids250, Global Land Evaporation Amsterdam Model (GLEAM), Global Lithologic Map (GLiM), and GLobal HYdrogeology MaPS (GLHYMPS). Additionally, a hydrologic disturbance index was created to indicate the most human-impacted catchments. Finally, we discuss the spatial variabilities of the attributes and their limitations of application.

## 2 The CABra dataset

 ## 2.1 Overview

The CABra dataset is a multi-source, multi-temporal, and multi-spatial resolution large-sample dataset for catchment attributes for Brazilian catchments. Using an extensive local/global high-quality data collection, we developed CABra considering eight main classes of attributes: topography, climate, streamflow, groundwater, soil, geology, land-cover, and hydrological disturbance. Gridded datasets of various kinds were averaged onto the selected catchments located over Brazil and neighboring countries, in the case of transboundary catchments. Moreover, we provide daily time series from climate and streamflow variables for a 30-year period, covering the hydrological years from 1980 to 2010, as described in Fig. 1.

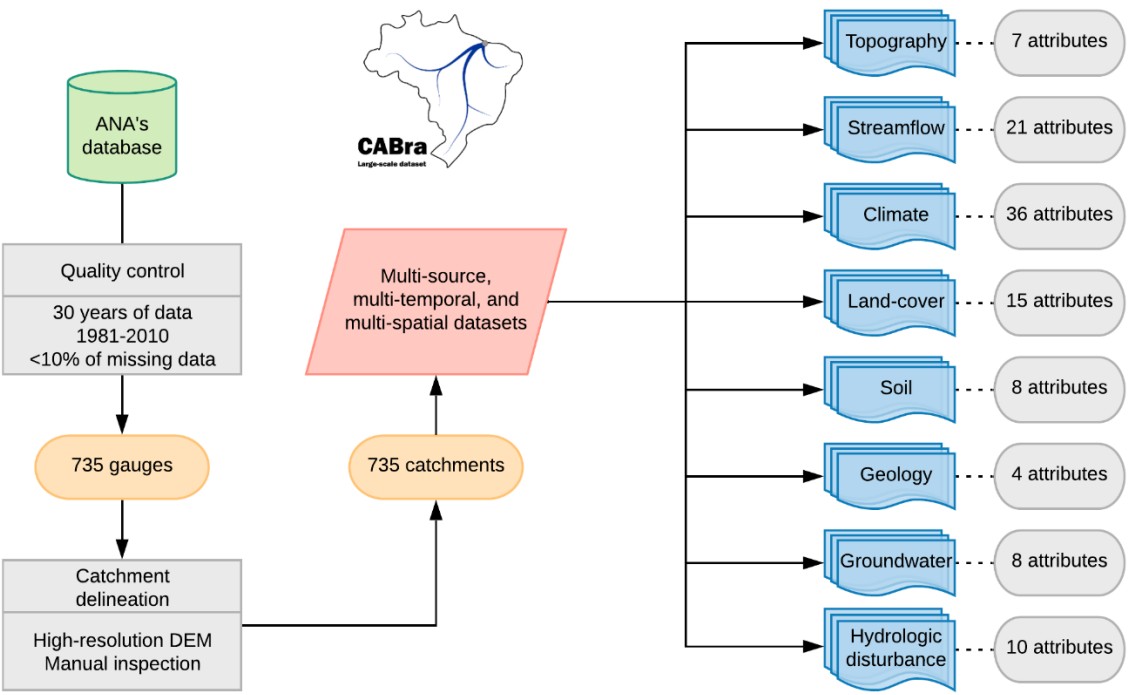

**Figure 1: Study delineation for the CABra dataset organization. From ANA's database, 735 were selected to integrate our dataset due to its high consistency and long time series of streamflow.**

The CABra dataset is recommended for a wide range of users for decision-making at multiple scales – local, national, or regional – covering all Brazilian biomes (Amazon, Cerrado, Atlantic Forest, Pantanal, Caatinga, and Pampa). CABra was created to ensure easy access to its information and provide high-quality data, with attributes useful for a variety of hydrometeorological modeling and assessments. Each catchment presents several attributes, ranging from the file information described in Table 1 to the attributes described throughout this article. Moreover, we made available all the geospatial data (shapefile of the boundaries) for the users.

**Table 1: General attributes of the CABra catchments.**

| Type | Attribute | Long name | Unit |
|---|---|---|---|
| **Identification** | cabra_id | CABra's identification code of the streamflow gauge | - |
| | ana_id | ANA's identification code of the streamflow gauge | - |
| **Location** | longitude | Longitude coordinate of the streamflow gauge | dd |
| | latitude | Latitude coordinate of the streamflow gauge | dd |
| | gauge_hreg | Brazilian hydrographic region of the streamflow gauge location | - |
| | gauge_biome | Brazilian biome of the streamflow gauge location | - |
| | gauge_state | Brazilian state of the streamflow gauge location | - |
| **Quality** | missing_data | Percentage of missing data | % |
| | series_length | Timeseries length of the streamflow gauge | years |
| | quality_index | Quality index of the CABra catchment records | - |

- Means dimensionless

## 2.2 Catchment delineation and topography

Brazil does not have an official database for the national catchments boundaries, and the Brazilian Water Agency (ANA) does not make available its geospatial database. Because of this and to avoid uncertainties in the existing datasets for South America, we freshly generated all the CABra catchments boundaries used in this study. Digital Elevation Model (DEM) quality and resolution are crucial at this stage since all the post-analysis with the multi-source information utilized in the CABra dataset are area-averaged. For example, is well-known that errors in topographic indices, e.g., slope and catchment area and boundary, are dependent on and highly sensitive to DEM resolution and accuracy, and it is suggested that, if available, a high-resolution DEM should be used instead of a low-resolution DEM due the negative effects of terrain generalization caused by them (Mukherjee et al., 2012; Vaze et al., 2010; Wechsler, 2007; Zhou and Liu, 2004). We delineated the CABra catchments following the procedure described in Maidment (2002), using streamflow gauges location information from the ANA's database and a high-resolution elevation product, i.e., the Multi-Error-Removed Improved-Terrain Digital Elevation Model with a 90-m spatial resolution at Equator (Yamazaki et al., 2017) (Fig. 2).

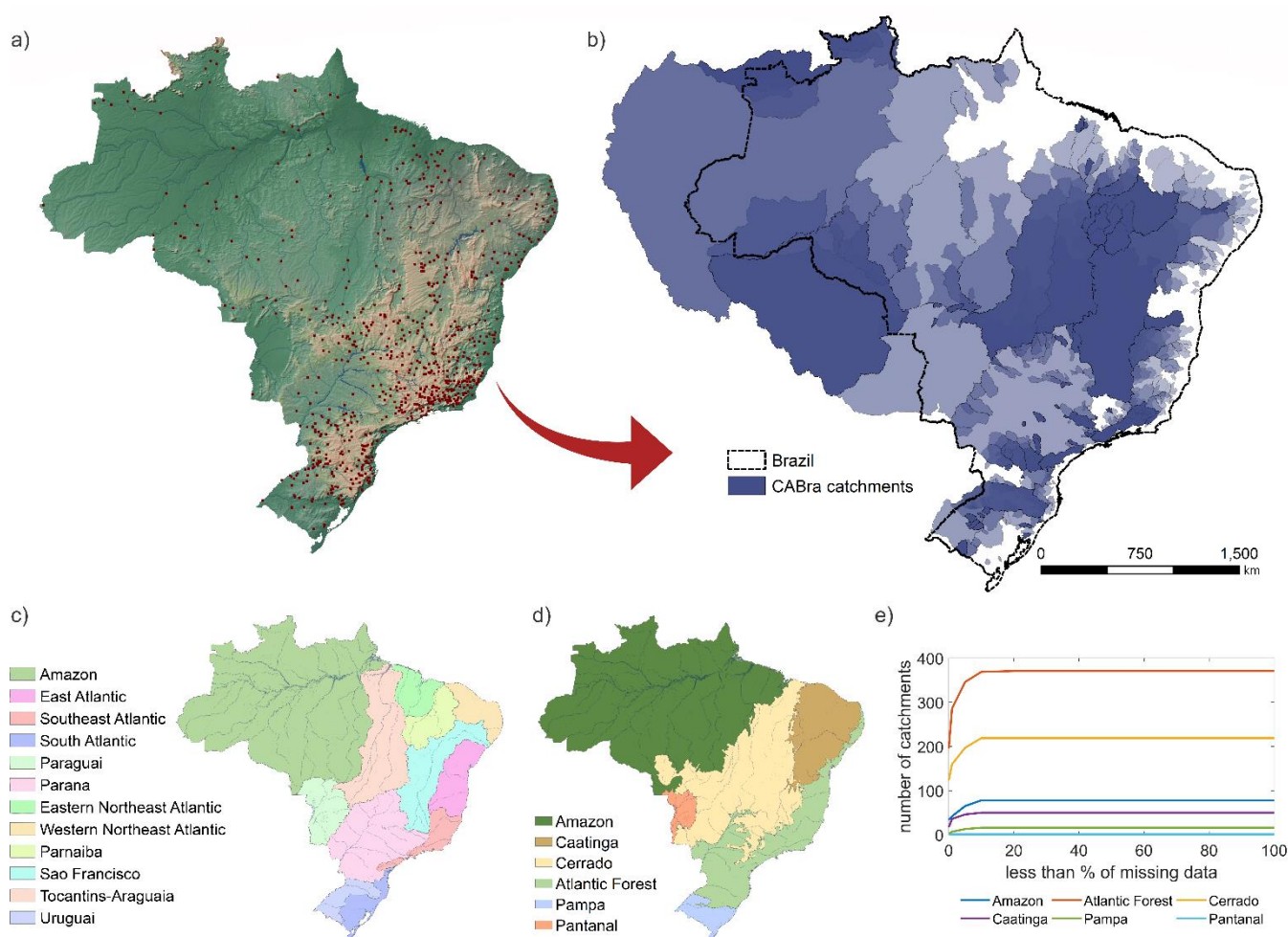

**Figure 2: Location map of the streamflow gauges and CABra catchments. a. Streamflow gauges coordinates of CABra catchments; b. The 735 CABra catchments boundaries; c. The 12 hydrographic regions of Brazil; d. The six main biomes of Brazil; e. Level of consistency of the streamflow gauges records for each biome.**

In the first stage, which we call "terrain processing", the DEM was sink-filled to avoid possible errors due to peaks or depressions. Then, the flow direction and flow accumulation were calculated, which indicates the direction and accumulation of flow, respectively, in each grid cell within the catchment. The next step was to define the stream network in the catchment. For the definition of a river stream, we considered a threshold of 100 cells accumulating water, and this value was chosen considering the DEM spatial resolution and the range of the size of the catchments. All the previous steps were run for the South America extension. Even though all outlets are located in the Brazilian territory, some of the drainage areas embrace larger areas outside of it. The second step was catchment delineation, where the products generated in the previous step and the coordinates of the streamflow gauges were used. Each streamflow gauge coordinate was first plotted as a point and the position of it to the stream network was checked and corrected, if necessary. The correction procedure was

performed for 132 out of CABra catchments. Then, each corrected point was used as an outlet of the catchment and the delineation of the drainage area was performed using the ArcHydro tool. Aside from the catchments limits, perimeters, and areas, we also extracted the stream information, such as the stream network and hierarchy (Strahler, 1952, 1957). It is important to highlight that we manually inspected each catchment outlet and area to overcome the limitation of unchecked boundaries of another existing catchment datasets, such as Do et al. (2018), which is based on a DEM with a spatial

resolution of 500-m. Moreover, this presented itself as a crucial procedure for an accurate delineation since several outlets' positions needed to be corrected to represent the real expected catchment boundary. Once the catchment boundaries were delimited, we calculated seven attributes related to the topography of each catchment: area, slope, maximum, minimum, and mean elevation, streamflow gauge elevation, and catchment order. The catchment boundaries and drainage network are also provided in CABra dataset.

**Table 2: Topography attributes of the CABra catchments.**

| Type | Attribute | Long name | Unit |
|------|-----------|-----------|------|
| **Elevation** | elev_mean | Mean elevation of the catchment | m |
| | elev_max | Maximum elevation of the catchment | m |
| | elev_min | Minimum elevation of the catchment | m |
| | elev_gauge | Elevation of the streamflow gauge | m |
| **Area** | catch_area | Area of the catchment | km² |
| **Slope** | catch_slope | Mean slope of the catchment | % |
| **Drainage** | catch_order | Strahler order of the catchment | - |

Figure 3 summarizes the topographic attributes for the CABra catchments. Catchment areas ranged from 9 to $4.8 \times 10^6$ km² (Fig. 3a). This large range of areas shows how Brazilian hydrology can be, at the same time, local and continental, necessitating a better understanding of hydrologic processes on different scales. Many of the largest catchments are in the

mainstream of one of the 12 hydrologic regions of Brazil, especially in the Amazon, Tocantins/Araguaia, São Francisco, Paraguay, and Paraná. The mean elevation of CABra catchments ranges from close to zero to up to 2000 m, with the highest values found in the southern and south-eastern portions. In turn, steepen areas can be found in the coastal and mountainous areas of the southeast and south (Fig. 3b and Fig. 3c). Most of the Brazilian catchments have a flat topography though, with a mean slope up to 10%. Figure 3d shows the gauge elevation. Note the difference between the gauge elevation and the mean

catchment elevation in Fig. 3b. The gauge elevation considers only the elevation at the gauge position in the landscape, thereby proving only the local information, while the mean catchment elevation considers the average elevation for the entire catchment. An example of this difference is the largest CABra catchment, i.e., the Amazon. The mean elevation in the Amazon basin would be low, however, the western part of the basin has some of the highest peaks of the Andes, where the gauge elevation would be much higher.

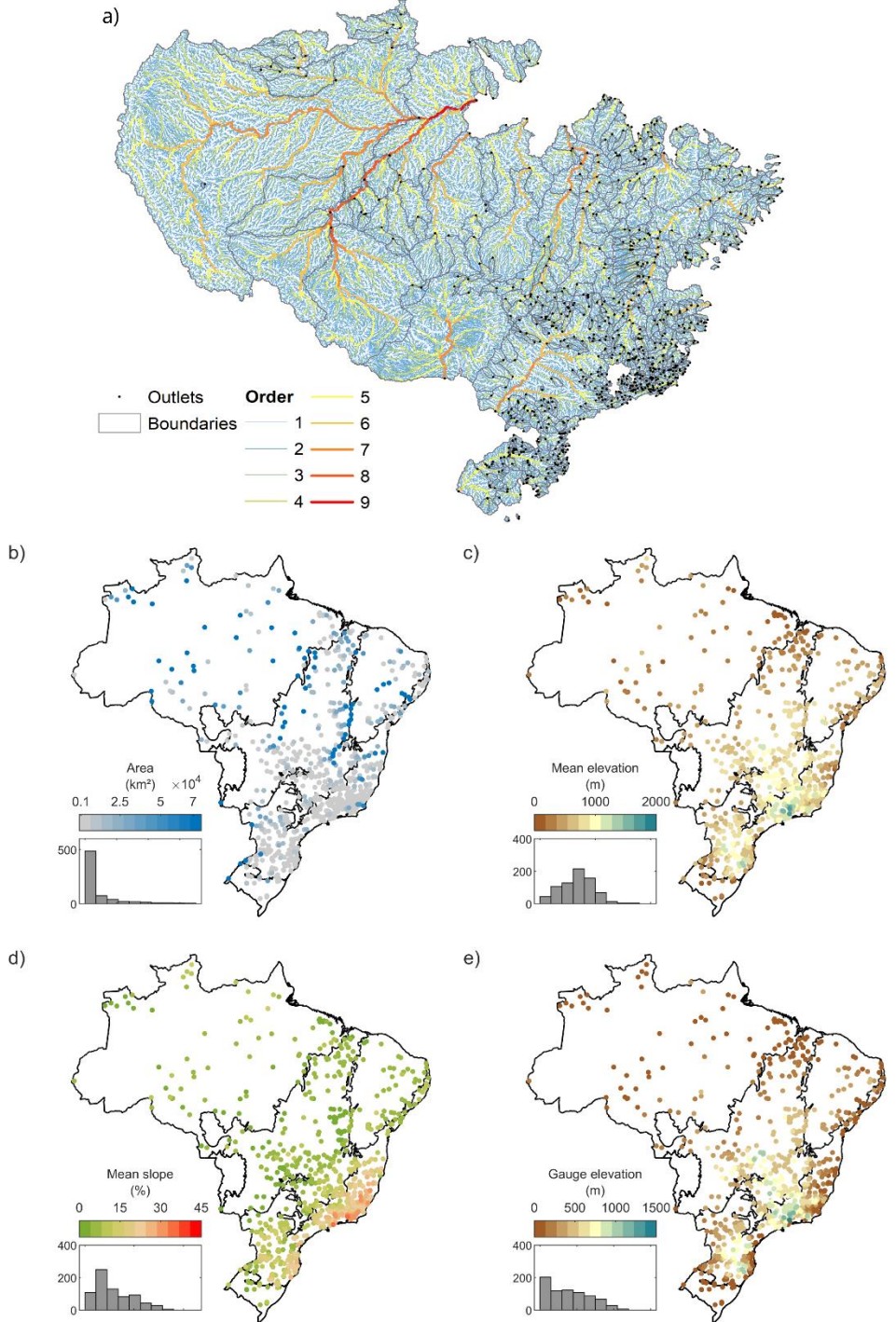


**Figure 3: Spatial distribution of the topography attributes of the CABra catchments. a. Stream order of Brazilian rivers; b. Area of the catchments, in km²; c. Mean elevation of the catchments, in m; d. Mean slope of the catchments, in percentage; e. Elevation of the streamflow gauge, in m.**

### 2.2.1 Uncertainty and limitations

The uncertainties related to the topography attributes are mainly related to the model terrain and streamflow gauges coordinates. The digital elevation model adopted for CABra catchments, developed by Yamazaki et al. (2017) is an improved product based on the composition of another baseline terrain products, such as the SRTM3 DEM, AW3D-30m DEM, and Viewfinder Panoramas DEM. Moreover, theres are gaps in high-relief mountains and water bodies that were filled manually for the final MERIT-DEM product, leading to 72% of mapped area with height accuracy better than 2 m

when slope < 10%. Regarding to streamflow gauges coordinates, there were inconsistences between the location provided by ANA and the stream network generated using the MERIT-DEM. We corrected the pair of coordinates, by matching the point to the nearest stream network, in a way that the area error against ANA's area was minized. Regarding to the catchment delineation, the uncertainty related to the automatic procedure conducted at the SIG environment is mainly dependent on the accucarcy, but some authors found that channels heads ($1^{st}$ order catchments) are the most subjected to greatest uncertainties

(Zandbergen, 2011).

### 2.3 Climate

### 2.3.1 Methodology

     We present daily time series of area-averaged precipitation, minimum, maximum, and mean temperatures, solar radiation, relative humidity, wind speed, evapotranspiration, and potential evapotranspiration (calculated by Penman-Monteith,

Priestley-Taylor, and Hargreaves methods). Moreover, we calculated several core climate indices, defined by the Climate and Ocean: Variability, Predictability, and Change project from the World Climate Research Programme (WCRP). Two main climate datasets were used in CABra. The first one, a high-resolution meteorological gridded dataset (0.25°x0.25°), developed by Xavier et al. (2016) (here referred to as "REF") is based on the spatial interpolation of meteorological data from ~4,000 rain gauges and wheatear stations in Brazil, from the ANA, Brazilian Institute for Meteorology (INMET, in

Portuguese), and Water and Power Department of São Paulo (DAEE/SP, in Portuguese), covering the period from 1980 to 2015. From these sets of meteorological gauges, 2890 are limited to precipitation data. This dataset is available at http://careyking.com/data-downloads/. This product has a much finer spatial resolution and is based on a higher number of rain gauge stations than other widely used products (~4,000 stations for Brazil, in comparison to ~600 stations for South America in CRU TS3.1 product). However, the REF dataset covers only the Brazilian territory, while the CABra dataset has

20 catchments with upstream areas outside Brazil. To overcome this, we incorporated the ERA5 (Hersbach et al., 2020) climate data into the CABra dataset (here referred to as "ERA5").

     ERA5 is the most recent version of climate reanalysis from the European Centre for Medium-Range Weather Forecasts (ECMWF) and provides hourly, daily, and monthly data on several atmospheric, sea, and land variables in a 0.25°x0.25° spatial resolution grid, from 1950 to the present. As a reanalysis dataset, the ERA5 uses past observations and models to

generate accurate and consistent time series of climate variables and parameters, being one of the widely used datasets in

geosciences (Hersbach et al., 2020). To incorporate and produce a more reliable product for all the CABra catchments, we have generated an ensemble mean product (here referred to as "ENS") using both datasets beforementioned, i.e., REF and ERA5 climate products. The procedure was conducted in the Climate Data Operators (CDO, Schulzweida, 2019) and aimed to a better characterization and representation of the climate based on the two independent estimations, which generally

imply in a more robust reproducibility of the phenomenon than in a single-member analysis (Abramowitz et al., 2018). Newman et al. (2015b) also found that ensemble product of precipitation and temperature still capture the main features of the variables and, moreover, improves the identification of extreme event frequency, and it is know that an ensemble usually outperforms individual forecasts (Bellucci et al., 2015; Solman et al., 2013; Tebaldi et al., 2005), being capable to detect internal variability and seasonal patterns. The ENS dataset generated here can be useful for climate-related analysis through

the Brazilian territory, since it merges two high-resolution and high-quality products.

The precipitation seasonality (Woods, 2009), which indicates the timing of the precipitation seasonal cycle and the temperature seasonal cycle – values close to +1 indicates summer precipitation and values close to -1 indicates winter precipitation – was calculated for the ensemble product.

The actual evapotranspiration adopted in CABra is derived from the Global Land Evaporation Amsterdam Model version 3

(GLEAM v3, Martens et al., 2017), which is a set of algorithms that estimates the many components of land evaporation based on satellite observations of climatic and environmental variables. The calculations of the actual evapotranspiration by GLEAM v3 take into account a potential evapotranspiration module (by Priestley and Taylor method), an interception loss module (by a Gash analytical model), and a stress module (by a semi-empirical relationship to root-zone moisture and vegetation optical depth). The GLEAM dataset is one of the most commonly used datasets on evapotranspiration

applications (Forzieri et al., 2018; Schumacher et al., 2019; Zhang et al., 2016).

Even though the REF dataset presents a reference evapotranspiration product (calculated by Penman-Monteith method following the FAO-56 guidelines), it embraces only the Brazilian territory and did not comprise all the areas of the catchments included in the CABra dataset. To overcome this limitation, we calculated the daily potential evapotranspiration (PET) by three different widely used methods based on energy balance and transfer mass, radiation, and temperature, using

meteorological variables from the ERA5 and the ensemble products as inputs. These three newly products are, for our knowledge, the most extent datasets of potential evapotranspiration for Brazil, covering a larger period than existent products, such the one introduced in Althoff et al. (2020) and Xavier et al. (2016).

The first method was the FAO-56 Penman-Monteith equation (Allen et al., 1998), which is the standard for reference evapotranspiration, and assumes a hypothetical crop similar to a surface of small grass of uniform grass, actively growing

and sufficiently watered. The FAO Penman-Monteith (PM) equation considers the energy budget and the aerodynamic and surface resistances of the crop and uses as inputs the solar radiation, air temperature, humidity, and 2m wind speed data (Equation 1).

$$PET_{PM} = \frac{0.408\Delta(R_n - G) + \gamma\frac{900}{T + 273}u_2(e_s - e_a)}{\Delta + \gamma(1 + 0.34u_2)}$$ 1

where $PET_{PM}$ is the reference evapotranspiration, in mm day$^{-1}$, $R_n$ is the net radiation, in MJ m$^{-2}$ day$^{-1}$, $G$ is the soil heat flux, in MJ m$^{-2}$ day$^{-1}$, $T$ is the mean daily temperature at 2m height, in ºC, $u_2$ is the wind speed at 2m height, in m s$^{-1}$, $e_s$ is saturation vapor pressure, in kPa, $e_a$ is the actual vapor pressure, in kPa, $\Delta$ is the slope vapor pressure curve, in kPa ºC$^{-1}$, and $\gamma$ is the psychrometric constant, in kPa ºC$^{-1}$.

The radiation-based method chosen for the CABra dataset is the Priestley-Taylor equation (PT) (Priestley and Taylor, 1972). The PT considers that when large areas, such as catchments, are saturated, the main force that governates the evaporation is the net radiation, and under certain conditions, the knowledge of net radiation and the ground dryness is enough to determine the vapor and sensible heat fluxes at the surface. Moreover, is one of the most commonly used models to estimate evapotranspiration due to its low number of inputs requirement (Maes et al., 2018; McMahon et al., 2013; Shuttleworth, 1996). The PT equation takes the following form:

$$PET_{PT} = \alpha\frac{\Delta}{\Delta + \gamma}(R_n - G)$$ 2

where $PET_{PT}$ is the potential evapotranspiration, in mm day$^{-1}$, $\alpha$ is the Priestley-Taylor constant, dimensionless, $R_n$ is the net radiation, in MJ m$^{-2}$ day$^{-1}$, $G$ is the soil heat flux, in MJ m$^{-2}$ day$^{-1}$, $\Delta$ is the slope vapor pressure curve, in kPa ºC$^{-1}$, and $\gamma$ is the psychrometric constant, in kPa ºC$^{-1}$. Considering that PT only considers daytime evapotranspiration and $G$ is negligible during the daytime, we used $G = 0$ in our calculations.

Priestley & Taylor (1972) empirically determined α for many locations and conditions in the world, ranging between 1.08 and 1.34. The authors concluded the best estimation for α should be an overall mean of 1.26. However, it is known that the α value is scenario-dependent and its variability is not taken into account when using the mean value proposed in its development (Guo et al., 2007).

The third method adopted here is the Hargreaves equation. The method was developed by Hargreaves (1975) for irrigation planning and design and it is a temperature-based equation widely used to calculate the potential evapotranspiration due to its easy application and low inputs requirement (Equation 3).

$$PET_{HG} = 0.0135\,R_s(T_a + 17.8)$$ 3

where $PET_{HG}$ is the potential evapotranspiration, in mm day$^{-1}$, $R_s$ is the solar radiation, in MJ m$^{-2}$ day$^{-1}$, and $T_a$ is the daily mean temperature, in ºC.

From the climatic variables and attributes, we carried out an analysis of the annual water balance in the Budyko space, an empirical approach applied to the study of the hydrological behavior of catchments. The Budyko hypothesis (Budyko, 1948, 1974) considers that the ratio between the long-term annual actual evapotranspiration (ET) and precipitation (P) is a function of the ratio between the long-term potential evapotranspiration (PET) and precipitation (P). The Budyko framework has been

235 used to assess global impacts of climate change on water resources (Berghuijs et al., 2017; Roderick et al., 2014), and to gain further insight on water balance controls at mean annual timescales (Donohue et al., 2007; Berghuijs et al., 2017; Meira Neto et al., 2020).

**Table 3: Daily series of meteorological variables and climate indices for the CABra catchments.**

| Type | Attribute | Long name | Unit |
|------|-----------|-----------|------|
| **Precipitation** | p_ref | Daily precipitation from the REF dataset | mm day$^{-1}$ |
| | p_era5 | Daily precipitation from the ERA5 dataset | mm day$^{-1}$ |
| | p_ens | Daily precipitation from the ENS dataset | mm day$^{-1}$ |
| **Temperature** | tmax_ref | Daily maximum temperature from the REF dataset | ºC |
| | tmin_ref | Daily minimum temperature from the REF dataset | ºC |
| | tmax_era5 | Daily maximum temperature from the ERA5 dataset | ºC |
| | tmin_era5 | Daily minimum temperature from ERA5 dataset | ºC |
| | tmax_ens | Daily maximum temperature from the ENS dataset | ºC |
| | tmin_ens | Daily minimum temperature from the ENS dataset | ºC |
| **Solar radiation** | srad_ref | Daily mean solar radiation from the REF dataset | MJ m² day$^{-1}$ |
| | srad_era | Daily mean solar radiation from the ERA5 dataset | MJ m² day$^{-1}$ |
| | srad_ens | Daily mean solar radiation from the ENS dataset | MJ m² day$^{-1}$ |
| **Wind** | wnd_ref | Daily mean 2m wind speed from the REF dataset | m s$^{-1}$ |
| | wnd_ era5 | Daily mean 2m wind speed from the ERA5 dataset | m s$^{-1}$ |
| | wnd_ ens | Daily mean 2m wind speed from the ENS dataset | m s$^{-1}$ |
| **Evaporation** | et_act | Daily actual evapotranspiration from the GLEAM v3 | mm day$^{-1}$ |
| | pet_pm | Daily potential evapotranspiration (Penman-Monteith method) | mm day$^{-1}$ |
| | pet_pt | Daily potential evapotranspiration (Priestley and Taylor method) | mm day$^{-1}$ |
| | pet_hg | Daily potential evapotranspiration (Hargreaves method) | mm day$^{-1}$ |
| **Climate Indices** | clim_p | Long-term mean daily precipitation (1980-2010) | mm day$^{-1}$ |
| | p_seasonality | Seasonality and timing of precipitation (1980-2010) | - |
| | clim_rh | Long-term mean daily relative humidity (1980-2010) | % |
| | clim_tmin | Long-term mean daily minimum temperature (1980-2010) | ºC |
| | clim_tmax | Long-term mean daily maximum temperature (1980-2010) | ºC |
| | clim_et | Long-term mean daily actual evapotranspiration (1980-2010) | mm day$^{-1}$ |
| | clim_pet | Long-term mean daily potential evapotranspiration (1980-2010) | mm day$^{-1}$ |
| | aridity_index | Aridity index (clim_p/clim_pet) of the catchment | - |
| | clim_srad | Long-term mean daily solar radiation (1980-2010) | MJ m² day$^{-1}$ |
| | clim_quality | Quality index of climate indices (indicates the source meteorological daily series used for long-term mean calculation) | - |

240 - Means dimensionless

## 2.3.2 Results and discussion

Figure 4 shows some of the climate attributes for the CABra dataset. Regarding the precipitation derived from our ensemble of Xavier et al. (2016) and ERA5 (Fig. 4a), we found the highest values, reaching up to 10 mm day$^{-1}$, in the northern portion, and the lowest values, below 1 mm day$^{-1}$, in the north-eastern portion. Despite the wide range in the daily precipitation, most of the catchments (~80%) presented area-averaged precipitation between 3 and 6 mm day$^{-1}$.

Figure 4d shows the area-averaged solar radiation reaching the surface, ranging from 10 to 20 MJ m$^2$ day$^{-1}$, with most of the catchments with daily values higher than 15 MJ m$^2$ day$^{-1}$. The spatial distribution of solar radiation is reflected in the temperature values in CABra catchments (Fig. 4e and Fig. 4f). The southern and south-eastern portions present the lowest values of both the maximum and minimum temperatures. This is due to the lower values of solar radiation and high altitudes found in these regions of Brazil. Other areas of Brazil are located in higher latitudes and are subject to higher solar radiation, and due to its flat relief, the temperatures are higher than in the south. Figure 4b indicates that, in most of CABra catchments (~85%), the precipitation seasonal cycle is in timing with the temperature seasonal dynamics, which means that most of the precipitation occurs in the summer (seas > 0). There are only a few catchments in the northern portion of Brazil that have precipitation in the winter (seas < 0), and this can be explained by the high influence of sea breeze on convective precipitation in this region. According to Ahrens (2010) and Kousky et al. (1984), the Amazonian coastal area is highly influenced by the sea breeze, which can occur in 3 out of every 4 days, with the formation of convective activity inland.

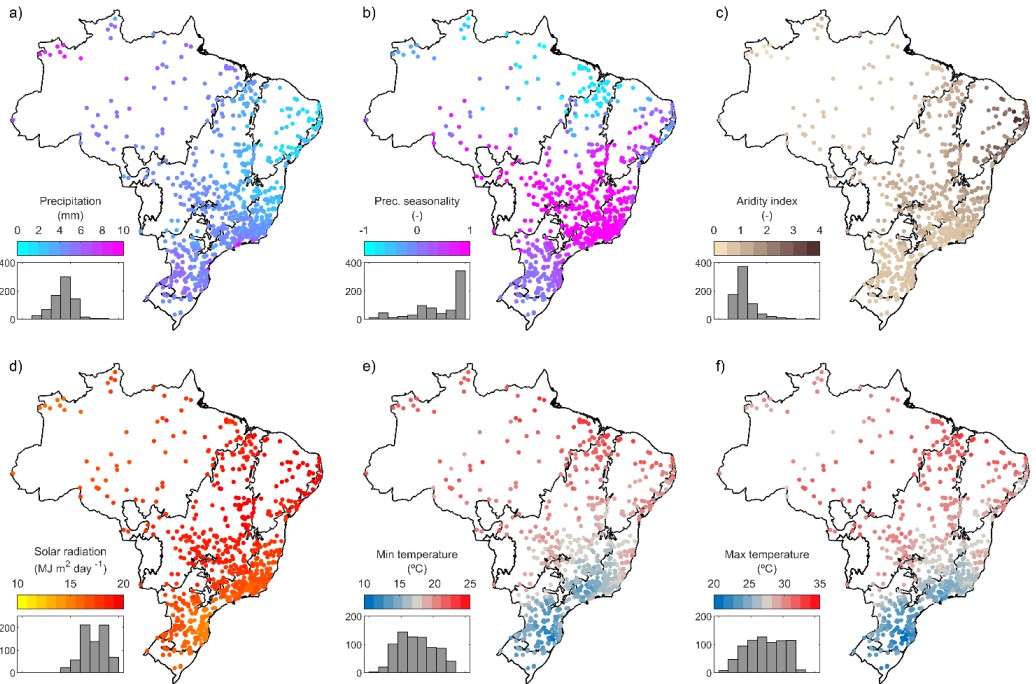

**Figure 4: Spatial distribution of climate indices of the CABra catchments. a. Mean daily precipitation, in mm day$^{-1}$; b. Precipitation seasonality, dimensionless; c. Aridity index, dimensionless; d. Mean daily solar radiation, in MJ m$^2$ day$^{-1}$; e. Mean daily minimum temperature, in ºC; f. Mean daily maximum temperature, in ºC.**

Our results of the computed potential evapotranspiration are presented in Fig. 5a, Fig. 5b, and Fig. 5c. They are related to three different methods for PET calculation, being: potential evapotranspiration for a reference crop using the Penman-Monteith equation; potential evapotranspiration by the Priestley-Taylor equation; and potential evapotranspiration by the Hargreaves equation. All the equations generated similar results of PET ranging from 3 to 6 mm day$^{-1}$, with similar spatial variability. The highest values were found for the north-eastern portion of Brazil, with the Penman-Monteith results being slightly higher than other equations. This could be related to the wind component in the method, which is not taken into account in the Priestley-Taylor and Hargreaves methods.

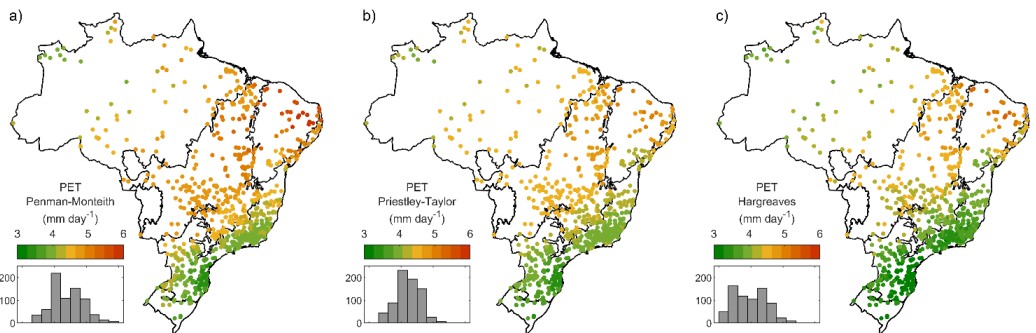

**Figure 5: Spatial distribution of the PET calculated from three different methods of the CABra catchments. a. Penman-Monteith method; b. Priestley and Taylor method; c. Hargreaves method.**

The Budyko framework (Budyko, 1948, 1974) shows that half of CABra catchments are water-limited and the other half are energy limited (Fig. 6). The lowest aridity index values are found in the Amazon and the Atlantic Forest, while the warmer and drier climate can be found in the Cerrado and Caatinga biomes. This may be correlated with the physiognomies of vegetation found in these biomes: tropical forests for the first group and grass and shrub for the second one, and especially, to the water availability and radiation incidence on these abovementioned biomes. Although we have found some outliers which are not explained by the Budyko hypothesis, most of the CABra catchments follow the expected behavior to the long-term mean water balance proposed by Budyko (1948, 1974). Moreover, we can note that the main climate features are captured by all the datasets, with catchments in Caatinga being more arid, followed by the Cerrado. The Atlantic Forest is in the same location at the Budyko space, while some catchments in Amazon only appears on ERA5 and ENS dataset, due to its extension outside REF. This shows the consistency between all datasets adopted in CABra.

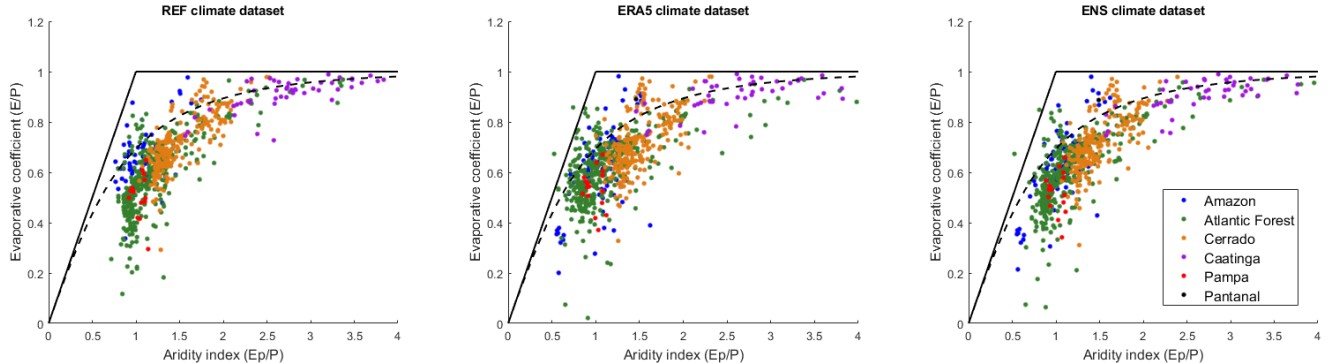

**Figure 6: Distribution of the CABra catchments in the Budyko framework from the three different climate dataset of CABra: REF, ERA5 and ENS. Values of E were estimated from the relation P = E + Q, considering long-term means.**

### 2.3.3 Uncertainty and limitations

The climate data provided by CABra dataset has limitations related to the number and spatial distribution of rainfall gauges in Brazilian territory that must be pointed. Since REF and ERA5 datasets are, respectively, ground-based and reanalysis gridded data, they are subject to uncertainties on the density of rainfall gauges network and in its post-processing procedures, which includes geospatial interpolation and data modelling and assimilation. In addition, REF dataset is not present in all of the 735 catchments due to its spatial extent, covering only the Brazilian territory. The quality of the data is presented for the users with a flag in the data though.

The potential evapotranspiration calculed for the CABra catchments are also subjected to uncertainties related to the equations chosen for the study and propagation of erros of input variables from climatic data. The golden standard for reference potential evapotranspiration ins the Penman-Monteith method, and the main limitations are related to the other two methods: on the application of the Pristley & Taylor method, the requirement of the Priestley-Taylor constant α, which is related to the ratio between the actual evapotranspiration and the equilibrium evaporation rate (Eichinger et al., 1996), is one of the greatest sources of uncertainty because it is scenario-dependent and its variability is not considered by using the mean value (α = 1.26) proposed in its development (Guo et al., 2007). On the other hand, the main limitation of Hargreaves equation for potential evapotranspiration is that the estimatons are subject to error due to a large range of temperatures caused by weather fronts on a daily scale. On the other hand, it is a less biased model, when compared to other methods, when applied to small and not well-watered catchments (Hargreaves and Allen, 2003).

## 2.4 Streamflow and hydrologic signatures

### 2.4.1 Methodology

The CABra dataset provides daily streamflow records for 735 catchments in Brazil. We used data from streamflow gauges of ANA, where each gauge is related to one of the abovementioned catchments. This dataset is available in the HIDROWEB database (see http://www.snirh.gov.br/hidroweb/). ANA's database contains raw time series of dozens of thousands of gauges of streamflow, precipitation, water quality, and sediment discharge, with a consistency level for each observation. Due to the inconsistencies and missing records in the streamflow data provided by ANA, we implemented filters to take into account only the reliable data for the CABra dataset.

During our analysis, we found four main issues with ANA's database collected from HIDROWEB: (a) missing streamflow values for a period of the time series; (b) duplicate streamflow values with different consistency levels; (c) duplicate values with the same consistency level, and (d) duplicate dates with different values and consistent levels. In the first filter step, we overcame the last three issues by picking up only one of the duplicated values/dates based on the best level of consistency. The first issue is more complex and difficult to overcome as in some cases the missing data reaches almost 100% for some gauges. Since long time series of streamflow is needed for reliable hydrologic investigations, we defined a threshold for the selection of the streamflow gauges considered in the CABra dataset based on the following conditions: at least 30 years of data, comprising the hydrologic years from 1980 to 2010, with up to 10% of missing data. The application of these filters led to 735 streamflow gauges, and consequently, 735 catchments. During the analysis, we also noted inconsistences on streamflow gauges data, such as extremely high values (up to 1,000 mm day$^{-1}$) and unexpected changes on daily streamflow values. Such inconsistences can lead to an under/overestimation of signatures based on mean values (e.g., mean daily flow, aridity index, runoff ratio) and, when repeated for a long time, it can modify signatures based on the frequency and dynamics of streamflow (e.g., flow duration curve, high and low flows frequency and duration). To avoid carrying these issues to the signatures calculation, we checked for outliers on the streamflow data by comparing each value to its neighbours. Elements with value larger than five times the median of a sliding ten-elements window (centred in 'x') were considered as an invalid value (NaN).

After the employment of the filters, we calculated for the 735 selected catchments, a variety of hydrological signatures, which can provide a better understanding of the patterns of functionality and behavior of the catchments. From the quantification of hydrological characteristics, it is possible to explain the variability in responses to climate forcings. We selected hydrological signatures obtained from widely available hydrological series (see Table 4), as well as Sawicz et al. (2011) e Westerberg e McMillan (2015). A list with more hydrological signatures can be found in Yadav et al. (2007). All the hydrological signatures were calculated considering the hydrological years (October 1$^{st}$ – September 30$^{th}$) from 1980 to 2010, as adopted by the Brazilian Water Agency in their annual reports (ANA, 2020a).

**Table 4: Hydrological signatures of the CABra dataset.**

| Type | Attribute | Long name | Unit |
|---|---|---|---|
| **Distribution** | q_mean | Mean daily streamflow | mm day$^{-1}$ |
| | q_1 | Very low streamflow (1$^{st}$ quantile) | mm day$^{-1}$ |
| | q_5 | Low streamflow (5$^{th}$ quantile) | mm day$^{-1}$ |
| | q_95 | High streamflow (95$^{th}$ quantile) | mm day$^{-1}$ |
| | q_99 | Very high streamflow (99$^{th}$ quantile) | mm day$^{-1}$ |
| **Frequency and duration** | q_hf | Frequency of high streamflow events | days y$^{-1}$ |
| | q_hd | Duration of high streamflow events | days |
| | q_lf | Frequency of low streamflow events | days y$^{-1}$ |
| | q_ld | Duration of low streamflow events | days |
| | q_hfd | Half-flow date | day of the year |
| | q_zero | Frequency of zero-flow events | days y$^{-1}$ |
| **Dynamics** | baseflow_index | Baseflow index | - |
| | q_cv | Coefficient of variation of daily streamflow | - |
| | q_lv | Coefficient of variation of low-flows | - |
| | q_hv | Coefficient of variation of high-flows | - |
| | q_elasticity | Elasticity of daily streamflow | - |
| | fdc_slope | Slope of flow duration curve (between 33$^{th}$ and 66$^{th}$ percentiles) | - |
| **Runoff** | runoff_coef | Runoff ratio | - |

- Means dimensionless

340

The hydrological signatures based on the distribution of the streamflow, we have used the daily streamflow and its quantiles to define the mean daily streamflow, very low-, low-, high-, and very high-flows. For the calculation of frequency and duration of the streamflow, besides the number of days with no flow, there was identified the number of days with 0.2 and 9 times the mean daily streamflow (low-flows and high-flows) and its number of days in sequency. The half-flow date

345 corresponds to the day of the year in which the cumulated annual streamflow reaches half of the annual totals. The baseflow index was calculated using a recursive digital filter proposed by Lyne and Hollick (1979), presented in Ladson et al. (2013). Additionally, regarding to the dynamics of streamflow, we calculated the coefficients of variation of the streamflow (mean, low, and high), the streamflow elascticity proposed by (Sankarasubramanian et al., 2001), which indicates the impact of changes in precipitation to the streamflow, and the slope of flow duration curve between 33$^{th}$ and 66$^{th}$ quantiles, which is a

350 good indicator of the perennial/non-perennial condition of the catchment. We also calculated the runoff coefficient for each

catchment, which indicates how much of the precipitated water becomes streamflow by the simple ratio between mean daily streamflow and mean daily precipitation.

### 2.4.2 Results and discussion

Figure 7 shows the hydrologic signatures calculated for the CABra catchments for the period between the hydrologic years 1980 and 2010. The mean daily flow for the Brazilian catchments ranges from less than 1 mm day$^{-1}$ to up to 9 mm day$^{-1}$, with an overall mean of 2 mm day$^{-1}$. The highest values were found in the extreme north of Amazon, where the daily flows reached 8 mm day$^{-1}$ due to high amounts of precipitation through the year, and in the Atlantic Forest, in the southeast, where we also have steepness relief with higher values of the slope, providing the runoff instead of infiltration process. This can be seen in Fig. 7b, related to the runoff coefficient, where we noted the high values in the southern and north-western portions of Brazil. Most of the CABra catchments presented a runoff coefficient up to 0.5 though.

Our results also revealed that the Brazilian catchments to be mainly dependent on the baseflow since all of it presented a baseflow index greater than 70%. The lowest values were found in the Caatinga biome, where we also found the lowest mean daily flows. The half-flow date (considering October 1$^{st}$ as the beginning of the hydrologic year) indicates that ~80% of Brazilian catchments reach the half of total accumulated annual flow in less than 200 days (Fig. 7d), showing the high correlation with the seasonal cycle of precipitation. The catchments with later dates of the half-flow day can be found in the Pampa biome, where there is no well-defined rainy/dry season, and in the Amazon, where the amounts of accumulated annual streamflow are too high and the peak of precipitation is near the end of the hydrologic year (Almagro et al., 2020). The analysis of the slope of the flow duration curve, in Fig. 7e, shows the lowest values in a great portion of Brazil, ranging from the Cerrado to the Atlantic Forest and Pampa biomes.

In our analyses, we also found zero values between the 33rd and 66th percentiles of the slope of flow duration curve in the north-eastern portion of Brazil, in the Caatinga biome, which indicates the existence of catchments with non-perennial rivers in that region, which are mainly dependent on direct runoff of rainfall. This can be also seen when analyzing Fig. 7f, related to the streamflow elasticity. The highest values, up to 4, are located in catchments within the same abovementioned region, indicating the strong dependence of those catchments on precipitation events to generate its streamflow. Moreover, we can note that most Brazilian catchments are inelastic to changes in precipitation. This fact can be explained by the high values of the baseflow index, which maintain the streamflow through the year. Fig. 7g, Fig. 7h, and Fig. 7i show the results related to the low flows of CABra catchments.

In general, Brazilian catchments present a low flow (5$^{th}$ quantile) lower than 1 mm day$^{-1}$, up to 50 days through the year, with a mean duration of up to 25 following days. Despite the mean values, we can note high values (up to 3 mm day$^{-1}$) in the Amazon. Additionally, higher values of frequency and duration of low flows can be found in the north-eastern portion of Brazil, with mean frequency reaching 150 days and mean duration reaching 100 days for some catchments. In turn, Fig. 7j, Fig. 7k, and Fig. 7l show the information about high flows in CABra catchments. Most CABra catchments present high flows up to 10 mm day$^{-1}$, but in some catchments, this value can reach 30 mm day$^{-1}$. As seen in the low flow analyses, the

mean frequency of high flow does not exceed 50 days per year for most of the catchments. The frequency, instead, lasts for
lower time, up to 10 days.

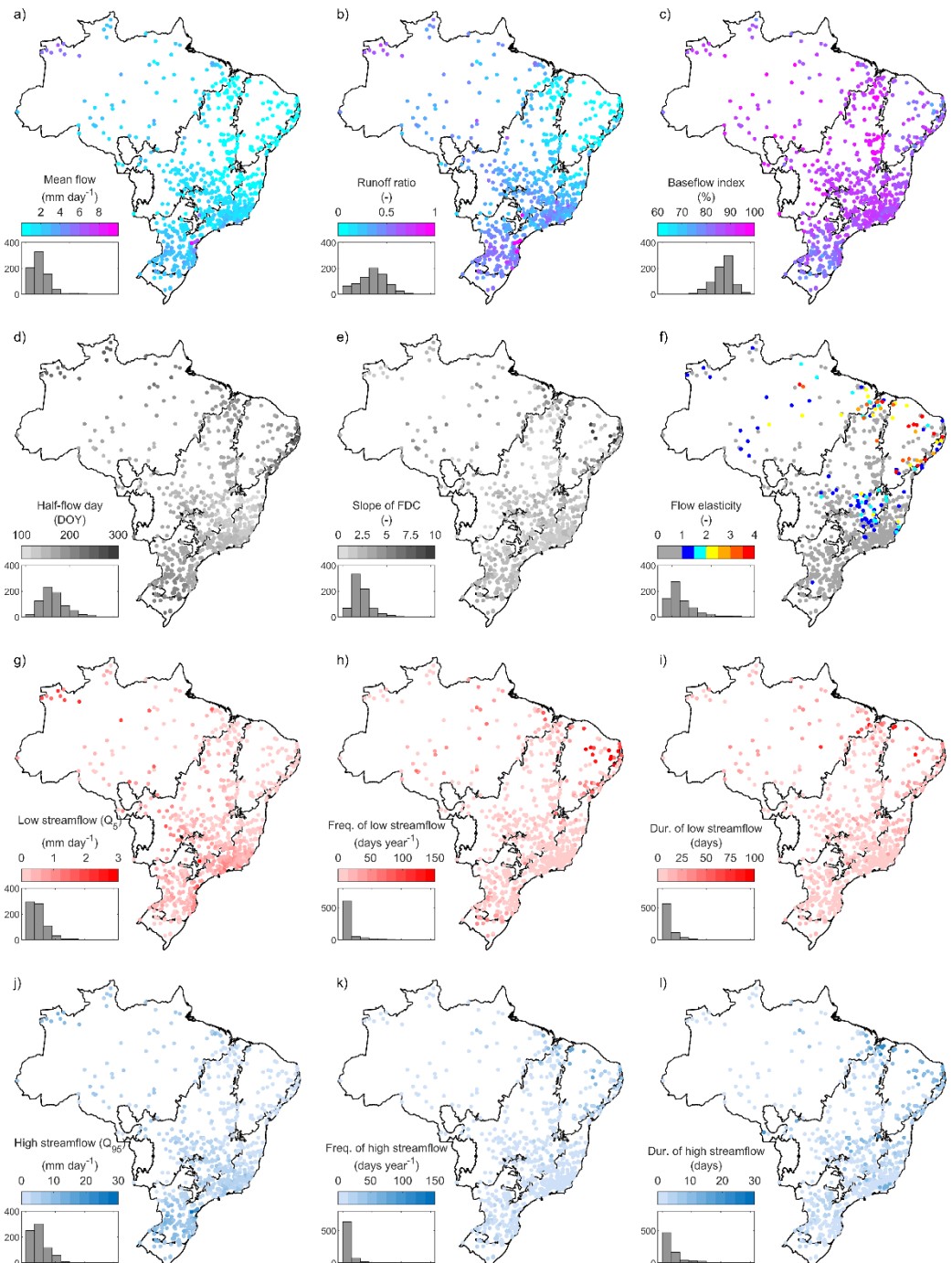

**Figure 7: Spatial distribution of the hydrological signatures of the CABra catchments. a. Mean daily streamflow, in mm day$^{-1}$; b. Runoff ratio, dimensionless; c. Baseflow index, dimensionless; d. half-flow day, in day of the year; e. The slope of the flow duration**

**curve, dimensionless; f. Elasticity of daily streamflow, dimensionless; g. Low streamflow, in mm day$^{-1}$; h. Frequency of low**
**streamflow events, in days year$^{-1}$; i. Duration of low streamflow events, in days; j. High streamflow, in mm day$^{-1}$; k. Frequency of high streamflow events, in days year$^{-1}$; l. Duration of high streamflow events, in days.**

### 2.4.3 Uncertainty and limitations

Uncertianties in the hydrologic signatures are mainly related to the daily streamflow data, which is, in turn, mainly related to the river discharge measurements and database maintence by the ANA. Data collection and stramflow measurements are not
the same in all catchments, varying from current meter to most advanced acoustic doppler profilers. The daily discharge of sections with well-stablished beds and long enough series of measurements are estimated by rating-curves, which are more susceptible to errors than direct measurements (Tomkins, 2014). Despite of this, daily streamflow records are provided with a consistence level, which can be "raw", meaning that data was not quality checked, or "consistent", meaning that data was quality checked. The consistence level is provided along with each daily record in CABra dataset, allowing the user to
identify best and worst periods of streamflow measurements in each catchment. Although it is impossible to accurate measure the uncertainties (as much as eliminate them) in a large-sample dataset such as CABra dataset, it is important to indicate the possible sources of them, since they are widespread in any hydrological modeling. This way we can indicate best periods for calibration/validation, increasing the reliability of the dataset and its application.

### 2.5 Groundwater

### 2.5.1 Methodology

The CABra dataset presents eigth attributes regarding the groundwater at the catchments (Table 5). They are related to the water table (water table depth and height above the nearest drainage) and to the aquifer where the catchment is within (aquifer name and rock type). The first attribute is the area-averaged water table depth. This information was extracted from Fan et al. (2013), which is a global water table depth map generated using a climate-sea-terrain coupled model. The results
were validated against observations and show the global patterns of shallow groundwater, making possible the understanding of how groundwater affects terrestrial ecosystems, such as the soil moisture and land hydrology, in a deficiency of rain (Fan et al., 2013; Lo et al., 2010).

The second attribute is the Height Above Nearest the Drainage (HAND), also related to the water table but is an indirect way to infer the water table depth. The HAND is a normalized drainage version of a digital elevation model, where the height is
defined as the vertical distance from a hillslope (at the surface cell) to a respective "outlet-to-the-drainage" cell, as defined by Nobre et al. (2011). Considering the local gravitational potential, the HAND model shows robust correlations between soil water conditions and its values. Additionally, the authors created three classes to easily infer about the water table depth (if at the surface, shallow or deep) only using a digital elevation model, which is commonly a piece of difficult and scarce information on a large scale. We also present the aquifer in which the catchment is within (most of the area) and the most
common type of rock of the aquifer. This information was provided by the ANA database and it is important to the

knowledge of the aquifer geology and its implication to the groundwater storage and recharge. We also have included data from experimental wells on the CABra catchments, when available. The data was provided by the Integrated Groundwater Monitoring Network (RIMAS) from the Geological Survey of Brazil (CPRM), and includes the location of each well and its static and dynamic levels.

**Table 5: Groundwater attributes of the CABra catchments.**

| Type | Attribute | Long name | Unit |
|------|-----------|-----------|------|
| **Water table** | catch_wtd | Water table depth | m |
| **Height above nearest drainage** | catch_hand | Height above the nearest drainage | m |
| | hand_class | Class of the height above the nearest drainage | - |
| **Aquifers** | aquif_name | Aquifer name | - |
| | aquif_type | Aquifer rock type | - |
| **Wells** | well_number | Number of experimental wells | - |
| | well_static | Static level of water table depth | m |
| | well_dynamic | Dynamic level of water table depth | m |

- Means dimensionless

### 2.5.2 Results and discussion

Our analyses showed a close relationship between the water table depth from Fan et al. (2013) and the HAND. In the northern portion of Brazil, especially in the Amazon, we can find shallow water table depths, while in the south-eastern, especially in the Atlantic Forest, we noted the deepest values for the water table depths (see Fig. 8a and Fig. 8b). This could be related to the altitudes of each catchment since the HAND is a product derived from a digital elevation model. As a catchment lies at a high elevation, the water table depth is deeper than the other catchments in low elevations. This is particularly noted in the coastal area of the Atlantic Forest, which presents high altitudes and at the same time, is close to the sea level. Values of water table depth and HAND are also in accordance to the experimental wells for catchments where this analysis were possible to carry. Despite this, the low density of experimental wells shows the lack of field data abour groundwater in Brazil.

Figure 8c shows that most of the CABra catchments are dominated by fractured and porous rocks. The fractured rocks store the water in fractures, creating large pockets of water. The porous rocks store water in the soil pores (especially in sandy soils originated by sedimentary rocks), and it is common to find large amounts of water in them. The two of the world's largest aquifers are in Brazil and are porous, the Guarani Aquifer in the Cerrado biome, and the Alter do Chão Aquifer in the Amazon biome. The third aquifer type found in CABra catchments is the karstic one. This can be found in the São Francisco River Basin.

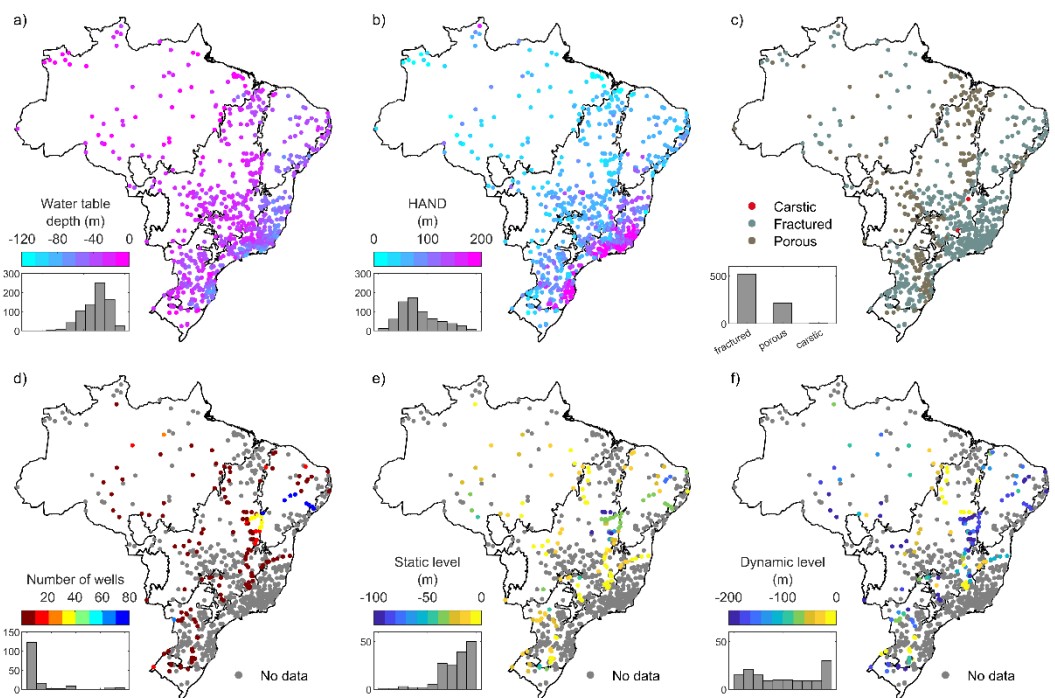

**Figure 8: Spatial distribution of the groundwater attributes of the CABra catchments. a. Water table depth, in m; b. Height Above Nearest Drainage, in m; c. Type of aquifer bedrock. d. Number of experimental wells; e. Static level, in m; f. Dynamic level, in m.**

### 2.5.3 Uncertainty and limitations

Due to the lack of a robust monitoring network for groundwater resources in Brazil, most of data Fan et al. (2013) for covering the Brazilian territory is based on in situ observations of water table depth and groundwater model forced by climate, terrain and sea levels, only up the 2013 year. For South America, there were 34,508 observation sites, most of them in Brazil, but they are concentred in the Atlantic coastal area, with few observations in most of Brazilian area. Moreover, the global dataset provided by Fan et al. (2013) neglects local perched aquifers, groundwater pumping, irrigation, drainage, and any other complexity of human interation. The HAND product, in turn, is not based on observations, but it is a simplified way to correlate the water table depth with terrain elevation, and it is mainly subject to errors in the digital elevation model used as input, especially in flat areas, where there are uncertainties during the flow direction determination (Nobre et al., 2011). The information of aquifers presented in the CABra dataset, provided by the Brazilian Water Agency, was developed with a previous and rigorous consistency analysis of geological and hydrogeological studies in Brazil, followed by the classification in three main classes, as fractured, carstic or porous. The mapping of aquifers systems was based on the analysis of consistency, adequacy and reclassification of existing geological and hydrogeological information. The reclassification of polygons from geological units and their groupings, according to their hydrogeological characteristics. Data sources with different scales, which might a uncertainty source for the aquifers data. The sources and spatial map of the aquifers is not available through CABra dataset, where we only present the most common aquifer in each catchment.

## 2.6 Soil

### 2.6.1 Methodology

The CABra dataset has eight attributes related to the soil type, properties, and texture (Table 6). The soil type of the catchment presented here is the most common type for each catchment (bigger percentage of the different types) derived from the Brazilian soil map developed by the Brazilian Agricultural Research Corporation (EMBRAPA, in Portuguese) (Santos et al., 2011). To meet with the international standards for soil classification, we converted the classes to the widely used World Reference Base (WRB) (FAO, 2014). Due to the high importance of the knowledge of the soil depth, density, texture, and organic matter to the understanding of soil-water dynamics and root grow (Dexter, 2004; Saxton et al., 1986; Saxton and Rawls, 2006; Shirazi and Boersma, 1984), we also present the mean areal attributes for them. These fields were taken from the SoilGrids250m, a global high-resolution gridded soil information based on field measurements, data assimilation, and machine learning. This is the most detailed and accurate global soil product and is crucial for the development of large-scale studies in many fields (ecology, climate, hydrology). However, despite all the improvements brought by SoilGrids250m, the data still have limitations, and one of the biggest is the high uncertainty levels for some of its products, such as the depth to bedrock and coarse fragments. Besides, we also employed the United States Department of Agriculture (USDA) soil texture classification, which is a widely used method for soil definition based on the mechanical limits of soil particles. Moreover, previous studies showed that the USDA soil texture classification can potentially reflect other soil parameters and characteristics (Groenendyk et al., 2015; Twarakavi et al., 2010), making it a powerful tool with a low input requirement.

**Table 6: Soil attributes of the CABra catchments.**

| Type | Attribute | Long name | Unit |
|---|---|---|---|
| **Soil type** | soil_type | Most common soil type | - |
| **Soil depth** | soil_depth | Soil depth to bedrock | m |
| **Soil density** | soil_bulkdensity | Soil bulk density | g cm$^{-3}$ |
| **Soil texture** | soil_sand | Sand portion on soil first layer | % |
| | soil_silt | Silt portion on soil first layer | % |
| | soil_clay | Clay portion on soil first layer | % |
| | soil_textclass | Soil texture classification (USDA) | - |
| **Organic content** | soil_carbon | Organic carbon content on soil first layer | ‰ |

- Means dimensionless

## 2.6.2 Results and discussion

The catchments presented 12 main soil classes, with the Ferrasols, Acrisols, and Nitisols being the most common soil types in more than 90% of the CABra catchments (Fig. 9a). The Ferrasols were the dominant soil type in approximately 75% of the catchments, typical of equatorial and tropical regions, which have an advanced stage of weathering of their constitutive material, being normally deep (>1m), well-drained, and acidic soils (high pH levels can occur in areas with a strong dry season, such as observed in the Caatinga biome). Acrisols are formed mainly by minerals, with an evident increase in the clay content from the surface to horizon B, with variable depth and drainage, but always with high acidity. The third most common soil type is the Nitisols, which have a clay texture, with a well-developed B horizon structure, and are usually deep and well-drained with moderate acidity (EMBRAPA, 2018).

We noted that most of the catchments present soil texture dominated by sand and clay (Fig. 9c, Fig. 9d, and Fig. 9e). South-eastern, northern, and central regions of Brazil are dominated by sandy clay loam soils, while the southern portion is dominated by clay, which can reach up to 80%, making this region one of the most productive in terms of agriculture in Brazil. By the employment of the USDA texture triangle, we found 6 classes: clay, clay loam, loam, sandy clay, sandy clay loam, and sandy loam (see Fig. 9b). The soils presenting a clay and clay loam texture are in the southern portion, especially where the Nitisols occur, which is also the region with a significant portion of the Brazilian agricultural production.

Most of the catchments present a mix of texture, the sandy clay loam, which covers from the south through the central to the northern regions of Brazil. There is a spatial correlation between the soil organic carbon, bulk density, and the distance to the bedrock, as we can see in Fig. 9f, Fig. 9g, and Fig. 9h. In the southern and south-eastern portions, especially in the Atlantic Forest biome, there is a combination of high soil organic carbon, low bulk density, and low distance to the bedrock. These characteristics, allied to the favourable climate, turned this region attractive to agriculture. On the other hand, other Brazilian regions present the opposite.

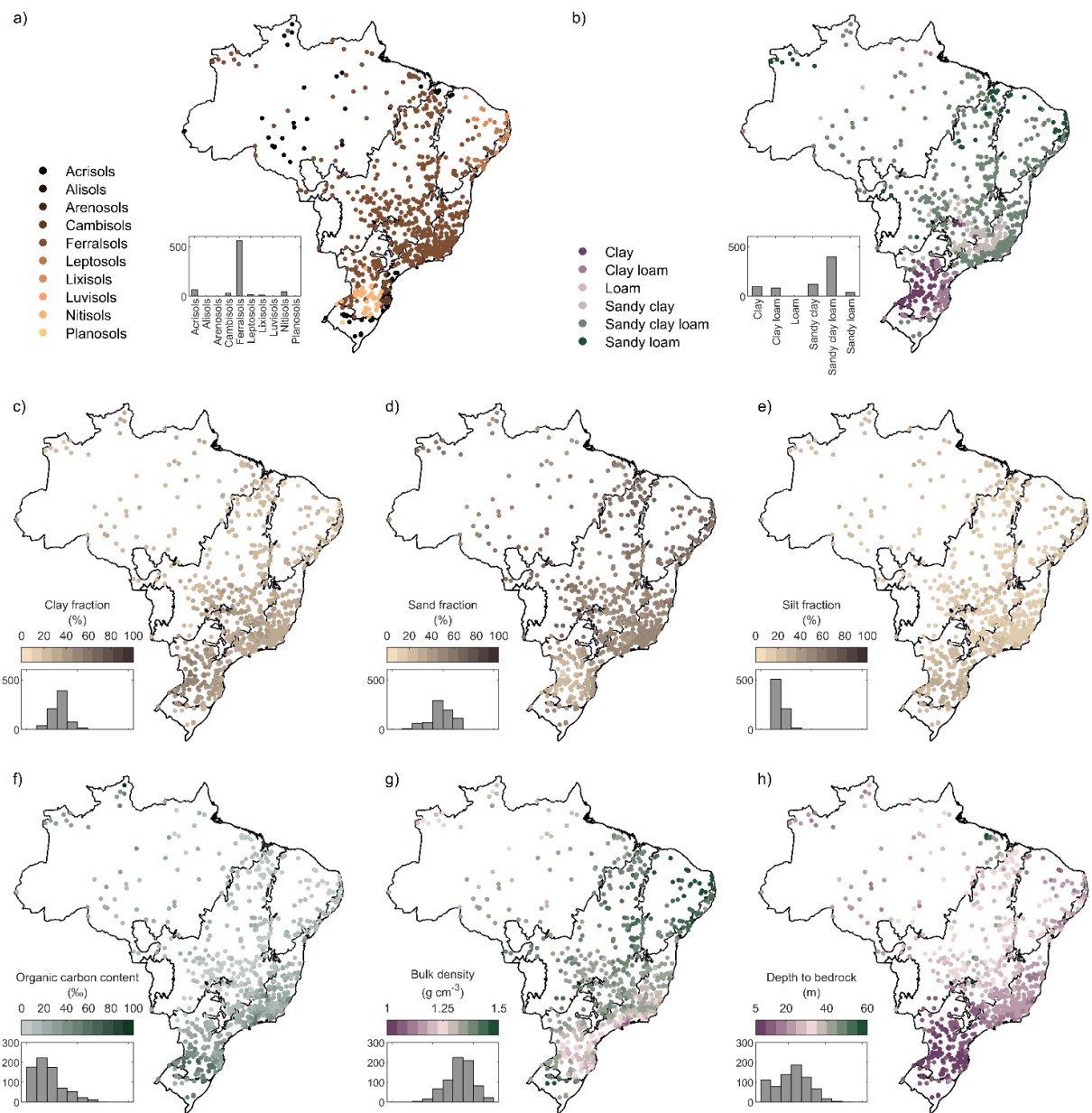

**Figure 9: Spatial distribution of the soil attributes of the CABra catchments. a. The most common type of soil in the catchment; b. The class of texture based on USDA classification; c. The clay fraction of the soil, in percentage; d. The sand fraction of the soil, in percentage; e. The silt fraction of the soil, in percentage; f. The organic carbon content of the soil, in permille; g. The bulk density of the soil, in g cm$^{-3}$; h. The depth to soil bedrock, in m.**

### 2.6.3 Uncertainty and limitations

The main limitation of the database used in CABra dataset as the source for soil attributes, the SoilGrids250 (Hengl et al., 2017), is related to the interpolation of a predicted data (through machine learning algorithms), which is based on soil profiles observed data. In this aspect, Brazil has a good starting point, with a dense and uniform distribution of in situ samples. However, authors state that, although most of properties are unbiased, coarse fragments and depth to bedrocks present relatively high uncertainties, as well overestimations in low values of organic carbon contente. Uncertainties are also related to the need of translation from the Brazilian classification system to the World Reference Base and USDA classification systems, where some information could be missed or misunderstood.

### 2.7 Geology

### 2.7.1 Methodology

The CABra dataset presents four attributes related to the geology of the catchments (Table 7), being the predominant lithology class, the porosity, the saturated permeability, and the saturated hydraulic conductivity. The lithology class is derived from the Global Lithologic Map (GLiM) (Hartmann and Moosdorf, 2012). The GLiM is a high-resolution global dataset that describes the geochemical, mineralogical, and physical properties of the rocks in 16 main lithological classes. Moreover, GLiM allows us to better understand the geology of smaller areas, such as our CABra catchments. Also, we are using a GLiM-derivate product of porosity and permeability named GLobal HYdrogeology MaPS (GLHYMPS), developed by Gleeson et al. (2014). The GLHYMPS is the first large-scale high-resolution mapping of porosity and permeability and fills a lack of robust and spatially distributed subsurface geology map.

The porosity is the void spaces in a material (soil in our case) controls how much fluid (water) can be stored in this material, or in the soil subsurface. The movement of the stored water in the soil is controlled by the permeability, which is the capacity of a porous material (again, soil) to transmit fluids. Both parameters are fundamental to the knowledge of fluid rate and its impacts on Earth's subsurface. When using this kind of high-resolution data for large-scale studies, we can improve our understanding of the dynamics between groundwater and land surface. Considering the saturated hydraulic conductivity as one of the most important physical properties on the quantitative and qualitative assessment of the water movement in the soil, we presented its values in the CABra dataset. Following the assumption that the hydraulic conductivity is separable into the contributions of the porous matrix of the soil, and the density and viscosity of the fluid, we also estimated the saturated hydraulic conductivity of the CABra catchments using its relation to the permeability (Equation 4), as described in Grant (2005).

$$K = \frac{k\rho g}{\mu}$$

where $K$ is the saturated hydraulic conductivity, $k$ is the saturated permeability, $\rho$ is the density of the fluid, $g$ is the gravitational constant (9.8 m s$^{-2}$), and $\mu$ is the viscosity of the fluid. In our study, we have considered the water as the fluid, so we have used $\rho$ = 999.97 kg m$^{-3}$, and $\mu$ = 0.001 kg m$^{-1}$ s$^{-1}$.

**Table 7: Geology attributes of CABra catchments.**

| Type | Attribute | Long name | Unit |
|---|---|---|---|
| **Lithology** | catch_lith | Most common lithology class | - |
| **Subsurface geology** | sub_porosity | Porosity | - |
| | sat_permeability | Saturated permeability | m² |
| | sat_hconduc | Saturated hydraulic conductivity | m s$^{-1}$ |

- Means dimensionless

## 2.7.2 Results and discussion

Related to the lithology class, the catchments present 10 different classes according to the GLiM dataset: siliciclastic sedimentary rocks, acid volcanic rocks, unconsolidated sediments, acid plutonic rocks, metamorphic rock, mixed sedimentary rocks, basic volcanic rocks, carbonate sedimentary rocks, intermediate volcanic rocks, and pyroclastic rocks (Fig. 10). We found that 35% of the catchments have the metamorphic rocks as the most common lithologic class, a result of continuous weathering on the original rock. These catchments are located especially in the southern portion of Brazil, in mountainous areas. Approximately 39% of CABra catchments are formed by sedimentary rocks, considering its subdivision in siliciclastic, unconsolidated, and mixed resulted from sediment deposition. They are mostly located in flat areas, such as in the Paraná River Basin and São Francisco River Basin, in the central and north-eastern portion of Brazil. 25% of catchments presents igneous rocks (plutonic and volcanic) as the most common lithology class, resulted from volcanic eruptions. These catchments are located mainly in the Atlantic Forest biome, although we can find some catchments in the Amazon.

In respect to the porosity, most CABra catchments presented values lower than 20%, with a mean value of 10%. Catchments in the Atlantic Forest presented the lowest values of the catchments set. Results regarding the saturated permeability and hydraulic conductivity reinforce the heterogeneity and random occurrence of these soil properties. As we can see in Fig. 10c and Fig. 10d, there is no well-defined spatial behavior for them. Saturated permeability ranges from -14 to -12 m² in log scale, with a mean of -13.4 m², while the saturated hydraulic conductivity presented a mean value of -6.4 m s$^{-1}$ in log scale, vary between -10 to -4 m s$^{-1}$ in log scale.

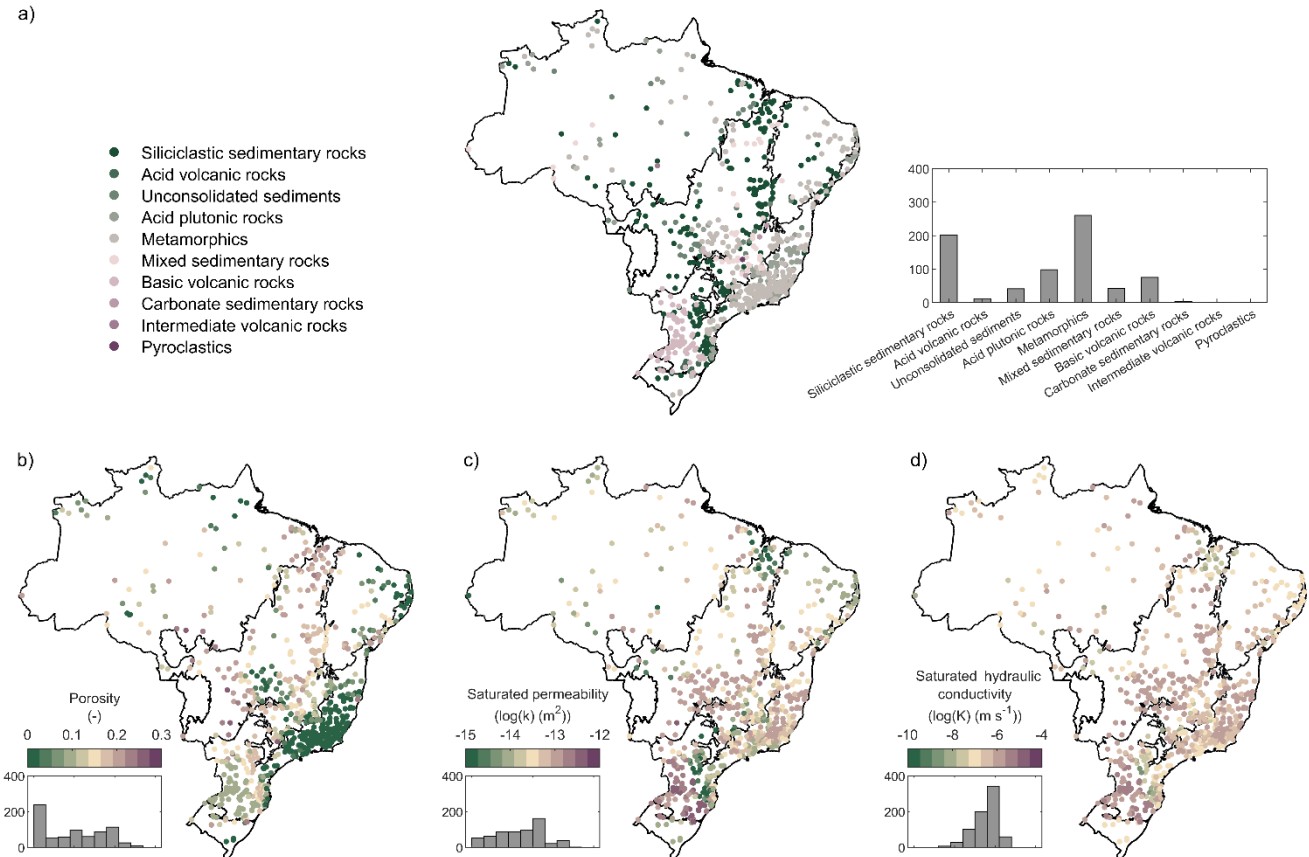

**Figure 10: Spatial distribution of geology attributes of the CABra catchments. a. Most common lithology class in the catchment; b. Porosity, dimensionless; c. Saturated permeability, in m$^2$; d. Saturated hydraulic conductivity, in m s$^{-1}$.**

### 2.7.3 Uncertainty and limitations

The geological map of the CABra dataset is derived from the GLiM dataset (Hartmann and Moosdorf, 2012), which is, in turn, the main source for the development of the hydrologeological map used in CABra dataset, the GLHYMPS (Gleeson et al., 2014). Authors state that the global lithological map is still subject to significant uncertainty in rock properties in some of its lithological classes, mainly because of the scale of the maps. About 14,6% of map's area are covered by mixed sediments, explicating the large amount of area subject to undistinguishable properties. In addition, quality of literature used to identify lithology in rare locations may have introduced some uncertainty level on GLiM. As mentioned before, the GLiM map was employed as a basemap for GLHYMPS permeability product, implying that all uncertainty associated to GLiM might be propagated to it. Moreover, Gleeson et al. (2014) presents a uncertainty map of permeability, showing high standard deviation values for central portions of Brazil, especially in Tocantins-Araguaia catchments. Finally, authors also recommend a careful use of the dataset where unsaturated zone processes are domintant, since GLHYMPS only takes in account saturated permeability.

## 2.8 Land-cover

### 2.8.1 Methodology

The CABra dataset presents 15 attributes regarding the land-cover and land-use of the Brazilian catchments (Table 8). They
are related to the area-averaged land-cover and land-use itself (dominant cover type, and the cover fractions of 9 main
classes of use: bare soil, forest, grass, shrub, moss, crops, urban, snow, and water) and to the area-averaged intra-annual
variability of the vegetation biomass, here represented by the Normalized Difference Vegetation Index. The land-cover and
land-use map used in the CABra dataset is the Copernicus Global Land Cover, which has 100-m spatial resolution, is a result
of a classification of the PROBA-V satellite observations of the year 2015 and follows the UN FAO Land Cover
Classification System (Buchhorn et al., 2019) available at https://land.copernicus.eu/global/lcviewer.

As an indicator for the vegetation biomass of the land-cover through the year, we are using the seasonal NDVI for each
CABra catchment. The NDVI is widely-used, easily accessible, and with high-temporal availability which can be useful for
many purposes on hydrology, since from as an annual precipitation cycle indicator to a input for soil erosion assessments.
We adopted a product derived from the Long Term Statistics (LTS) based on the Normalized Difference Vegetation Index
(NDVI) from the Copernicus Global Land services. This dataset is an NDVI mean for each month of the year during the
1999-2017 period, obtained from the SPOT-VGT and PROBA-V sensors in a 1-km spatial resolution, available at
https://land.copernicus.eu/global/products/ndvi. The NDVI is obtained by calculating the spectral reflectance difference
between red and near-infrared bands of the satellite image (Tucker, 1979) (Equation 5) and ranges from -1 to +1, with the
highest values attributed to areas with greater vegetation cover.

$$\text{NDVI} = \left(\frac{\text{NIR} - \text{RED}}{\text{NIR} + \text{RED}}\right)$$

where NIR is the surface spectral reflectance in the near-infrared band and RED is the surface spectral reflectance in the red
band.

**Table 8: Land-cover attributes of CABra catchments.**

| Type | Attribute | Long name | Unit |
|------|-----------|-----------|------|
| **Land-cover and land-use** | cover_main | Dominant cover type | - |
| | cover_bare | Bare soil fraction of cover | % |
| | cover_forest | Forest fraction of cover | % |
| | cover_grass | Grass fraction of cover | % |
| | cover_shrub | Shrub fraction of cover | % |
| | cover_moss | Moss fraction of cover | % |
| | cover_crops | Crops fraction of cover | % |
| | cover_urban | Urban fraction of cover | % |
| | cover_snow | Snow fraction of cover | % |
| | cover_waterp | Water fraction of cover (permanent) | % |
| | cover_waters | Water fraction of cover (seasonal) | % |
| **Vegetation** | ndvi_djf | DJF normalized difference vegetation index | - |
| | ndvi_mam | MAM normalized difference vegetation index | - |
| | ndvi_jja | JJA normalized difference vegetation index | - |
| | ndvi_son | SON normalized difference vegetation index | - |

- Means dimensionless

### 2.8.2 Results and discussion

We observed that most of the Brazilian catchments are covered by forest and grassland (Fig. 11). The shrub is the dominant cover for most of Caatinga catchments, while the grass is the dominant one in the Cerrado (tropical savannah). The forest cover is dominant especially in the Amazon and Atlantic Forest, as these two biomes are known by tropical forest occurrence, but even though the forest cover is not the most common for all the CABra catchments, ~85% of them present at least 20% of it (Fig. 11b). The grass cover fraction presented values up to 40% of the area for most of the catchments but reached 60% in some cases (Fig. 11c). The highest values were found in the Cerrado and Atlantic Forest biomes, in central and south-eastern portions of Brazil.

Large areas of natural cover were converted to agricultural lands (including crops and pasture) in past years (Gibbs et al., 2010, 2014), and satellite sensors and classifiers algorithms cannot separate natural grassland and pasture/managed grasslands, as described in the PROBA-V documentation. Figure 11d gives us a better idea of this. Probably the fraction of the shrub cover of the Cerrado is the natural cover remaining for this biome since this is the expected type of vegetation. As

seen in Fig. 11e, a few numbers of catchments present the crops as the dominant cover type, mostly in the central and

southern region, but we can also see the great fraction of crop cover in the MATOPIBA region, one of the largest agriculture

frontiers in Brazil (Gibbs et al., 2014; Pires et al., 2016; Spera et al., 2016). Figure 11f shows that there are only a few cases

of urban catchments, within or close to major Brazilian cities that present this type of cover, showing that the CABra dataset

is mainly composed of either natural or minimally (hydrologically) modified catchments.

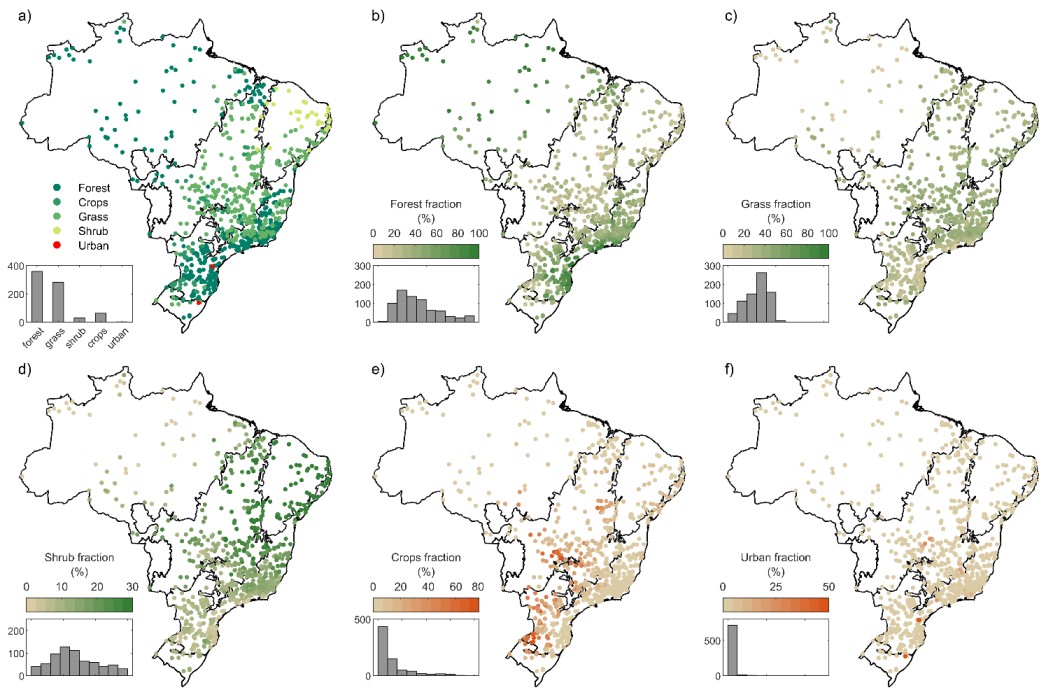

**Figure 11: Spatial distribution of the land-cover and land-use attributes of the CABra catchments. a. The most common land-cover type in the catchment; b. Forest fraction of land-cover, in percentage; c. Grass fraction of land-cover, in percentage; d. Shrub fraction of land-cover, in percentage; e. Crops fraction of land-cover, in percentage; f. Urban fraction of land-cover, in percentage.**



The seasonal variability of the NDVI can be seen in Fig. 12. Although the mean seasonal values for the entire country are

similar (0.65 for DJF, 0.69 for MAM, 0.64 for JJA, and 0.56 for SON), the spatial variability of the NDVI values are

noticeable. There is a clear relationship with the annual cycle of precipitation, and that is why it is so important to consider

the seasons to analyze the NDVI. Higher values of NDVI occurs in the accordance to the seasonal cycle of precipitation in

all the biomes, especially in DJF and MAM months. Even in the Amazon, we can see a considerable decrease in the NDVI

values for the catchments in the dry seasons (JJA and SON) as well as the other biomes and regions of Brazil. NDVI reaches

the lowest values at the end of the hydrological year and then starts to increase the values only at the beginning of the rainy

season, i.e., DJF season. Intermediate values in the central portion of Brazil are much likely to be linked to agricultural

production, leading the values to be lower than the natural cover.

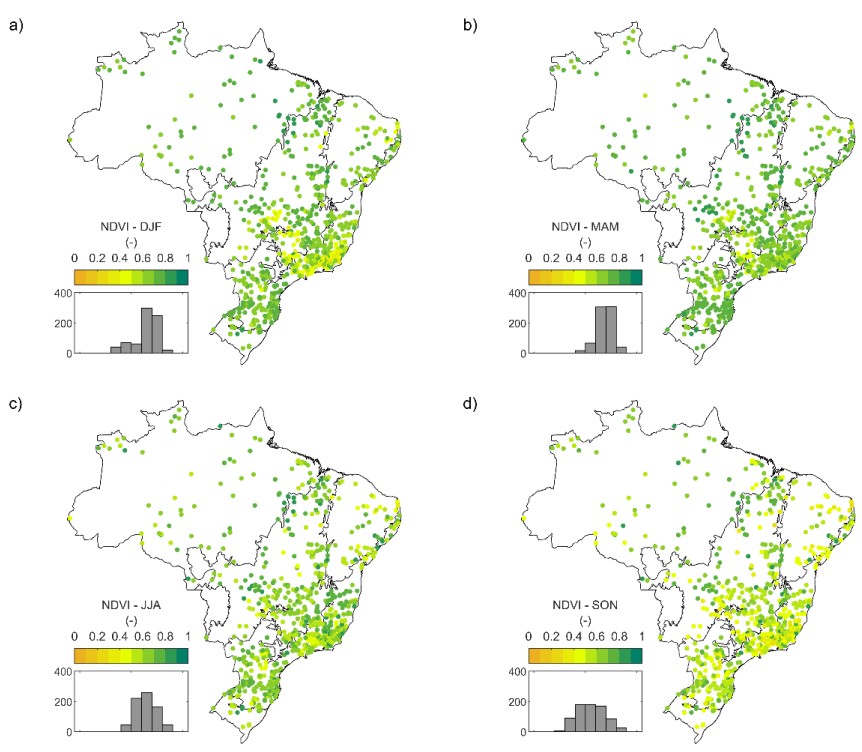

**Figure 12: Spatial distribution of the seasonal NDVI of the CABra catchments. a. NDVI in summer season (DJF); b. NDVI in autumn season (MAM); c. NDVI in the winter season (JJA); d. NDVI in the spring season (SON).**

### 2.8.3 Uncertainty and limitations

Although the CABra dataset presents one of the most high-accucary spatial resolutions in a global scale, the data is related to the 2015 year, which is not within the 1980-2010 period adopted in the hydrological analyses.

As authors from the Copernicus Global Land Cover (Buchhorn et al., 2019) states, the global land-cover data should be used with confidence but with careful and critical analysis by the users, due to the land changes commissions and omissions. Uncertainty analyses conducted in three aggregated classes (forest, crops and natural vegetation) showed high accuracy in all regions of the world, when compared with more than 200,000 samples points. Eventhough, there is some level of overestimation in the forest class, leading to a careful assessment of land-cover in Amazon and Atlantic Forest catchments. At the same time, due to the 100 m spatial resolution, small villages and highly fragmented landscapes might be indistinguishable and/or mixed with different classes.

NDVI dataset, also provided by Copernicus Global Land Cover, should be used as a qualitative indication of the biomass in the catchment, due to it relatively low spatial resolution (300 m). There are also uncertainties related to the radiometric calibration of the images, anisotropic surfaces, aside from the fact of the products did not considered adjacency effects and slope correction.

## 2.9 Hydrologic disturbance

### 2.9.1 Methodology

The CABra dataset presents 10 attributes related to the hydrologic disturbances on catchments water fluxes (Table 9). Anthropic changes in water flux patterns, which happens outside the range of natural flow and climate extremes, can directly impact the water availability and quality, stream channel geometry and sedimentation, and the equilibrium of ecosystems (Boulton et al., 1992; Coleman et al., 2011; Whited et al., 2007). Natural conditions of catchments are constantly modified by human interactions such as land-cover and land-use changes, flow regulation, water abstractions, soil impermeabilization, and many others, which can drastically alter the way hydrologic fluxes in the catchments respond. Then, our goal was to create a simple index, with easily accessible inputs, that is capable to measure how much disturbed a catchment is in relation to its hydrology. Since the beginning of CABra development, it was known that most of the catchments were minimally urbanised, but with some of them with changes in the original land-cover (conversion of natural vegetation to cropland/pasture). Some studies conducted in Brazil found that, besides the fact of the interference by the conversion of natural vegetation to pasture, this led to minimal changes in the surface hydrology of the catchment, being more relevant to groundwater recharge and soil chemistry (Bacellar, 2005; Lanza, 2015; Nepstad et al., 1994; Salemi et al., 2012). Additionally, it has been seen that the human-induced impact of the reservoirs can be more relevant than the natural ones, and can significantly alter natural hydrological processes (Zhao et al., 2016), leading to an increase/decrease of streamflow and hydrological droughts characteristics (Wanders and Wada, 2015; Ye et al., 2003; Zhang et al., 2015). Moreover, Zhang et al. (2015) found that hydrologic vulnerability is also directly related to human water abstractions, but this can be compensated by streamflow regulation of the reservoirs. This led us to an integrated analysis of the reservoir regulation and human water abstract to reach the optimal balance on our index.

Based on the abovementioned, we have decided to use weighted information about the land-cover, reservoirs, and water demand of each catchment. We considered the reservoir-based information with more impact: regulation capacity with 40%, number of reservoirs and its percentage of catchment area with 5% each. The second most impacting factor of the index is the non-natural land-cover in the catchment, which can lead to modify hydrological surface and subsurface processes, with 40% of the weights. Finally, the water abstraction of the catchment was pondered with 10%.

In the development of this index, we have considered fraction of urban cover in each catchment, the distance to the nearest urban area of each catchment (considering any pixel of urban area), the number of reservoirs in each catchment (ANA, 2020b), the total volume of reservoirs in each catchment (ANA, 2020b), and its flow regulation capacity, the fraction of reservoir area of each catchment area (ANA, 2020b), and the annual water demand (ANA, 2019b). The equation related to the hydrologic disturbance index can be found in the following Equation 6:

$$HD_{index} = 0.4([U_C.U_D] + CR_C) + 0.05R_N + 0.05R_{\%A} + 0.4R_R + 0.1W_D \qquad 6$$

where $HD_{index}$ is the hydrologic disturbance index, dimensionless; $U_C$ is the normalized fraction of urban cover; $U_D$ is the normalized distance to the nearest urban area; $CR_C$ is the normalized fraction of crops cover; $R_N$ is the normalized number of reservoirs; $R_{\%A}$ is the normalized percentage of catchment's area covered by reservoirs; $R_R$ is the normalized reservoirs' regulation capacity of catchment's mean annual flow; and $W_D$ is the normalized catchment's annual water demand.

**Table 9: Hydrologic disturbance attributes of CABra catchments.**

| Type | Attribute | Long name | Unit |
|---|---|---|---|
| **Reservoirs** | res_number | Number of catchment's reservoirs | - |
| | res_area | Total area of catchment's reservoirs | km² |
| | res_area_% | Catchment's area percentage covered by reservoirs | % |
| | res_volume | Total volume of catchment's reservoirs | hm³ |
| | res_regulation | Reservoir's regulation capacity of the mean annual flow | - |
| **Water demand** | water_demand | Water demand in the catchment | mm year⁻¹ |
| **Land-cover** | cover_urban | Urban fraction of cover | % |
| | cover_crops | Crops fraction of cover | % |
| | dist_urban | Distance from gauge to nearest urban cover | km |
| **Hydrologic disturbance index** | hdisturb_index | Index of hydrologic disturbance in the catchment | - |

- Means dimensionless

The result is the hydrologic disturbance index (HDI), which will easily provide for CABra users the degree of human interactions that can modify water fluxes in each catchment. Additionally, we also applied a random forest algorithm for a regression analysis to show if and how the hydrological signatures are captured by the HDI.

**2.9.2 Results and discussion**

The results of the spatial distribution of the hydrological disturbance index and its components are shown in Fig. 13. Most CABra catchments are close to an urban cover (it can be a large city or a small village), with a distance of up to 10 km. However, we also could find catchments with up to 100 km of distance to the urban cover. As seen in Fig. 13b and Fig. 13c, most CABra catchments present a fraction of urban cover up to 10%, with high values close to large cities, and a fraction of crops cover up to 40%, with the highest values in central and southern portions. As these factors present a high weight on the hydrological disturbance index, they are a good clue of the most disturbed catchments.

Results from the reservoirs in CABra catchments are shown in Fig. 13d, Fig. 13e, Fig. 13f, and Fig. 13g. The number of reservoirs in the catchment ranges from zero to 48,404. Even though we found the largest number of reservoirs in a large catchment, this relationship is not linear. There are some catchments, especially in the São Francisco River Basin, which

presents an extremely high number of reservoirs due to the low amounts of annual precipitation and intensive drought in the region. Moreover, catchments in the São Francisco River Basin presents the highest values of the total volume of reservoirs.

These reservoirs are used for many anthropogenic purposes, such as hydroelectric power plants, irrigation, drinking water supply, fish-farming, and recreation. These high values of the total volume of reservoirs, especially in the drier regions, could lead to a strong streamflow regulation, as seen in Fig. 13g. In most of the CABra catchments, reservoirs can regulate up to 25% of the annual flow, but there are some cases in the Caatinga biome where the regulation capacity reaches up to ten times the annual flow, making these catchments susceptible to non-natural events.

The water demand on CABra catchments ranges from zero (in Amazon) to 171 mm year$^{-1}$ (in Caatinga) and it is related to drinking water supply and irrigation of agricultural areas (Fig. 13h). The integrated analysis of the above-mentioned attributes is shown in Fig. 13i, as the new hydrological disturbance index. Most of the CABra catchments present an index value of up to 0.2, indicating a low anthropic interference on water fluxes. Higher values, above 0.4, indicate catchments with some significant interference on water fluxes, which may be related to one or more terms of the equation. High values

of the hydrological disturbance index in the central and southern portion of Brazil may be related to agriculture development, while in the south-eastern part, they may be related to urbanization, and in the north-eastern part, they may be related to the presence of numerous voluminous reservoirs. As expected, in the Amazon and mountainous areas of Atlantic Forest, low values were found. The creation of the hydrological disturbance index can be especially useful for the users of the CABra dataset, allowing them to quickly view the general state of the anthropogenic interferences on water fluxes, which is an

important consideration in a wide range of studies.

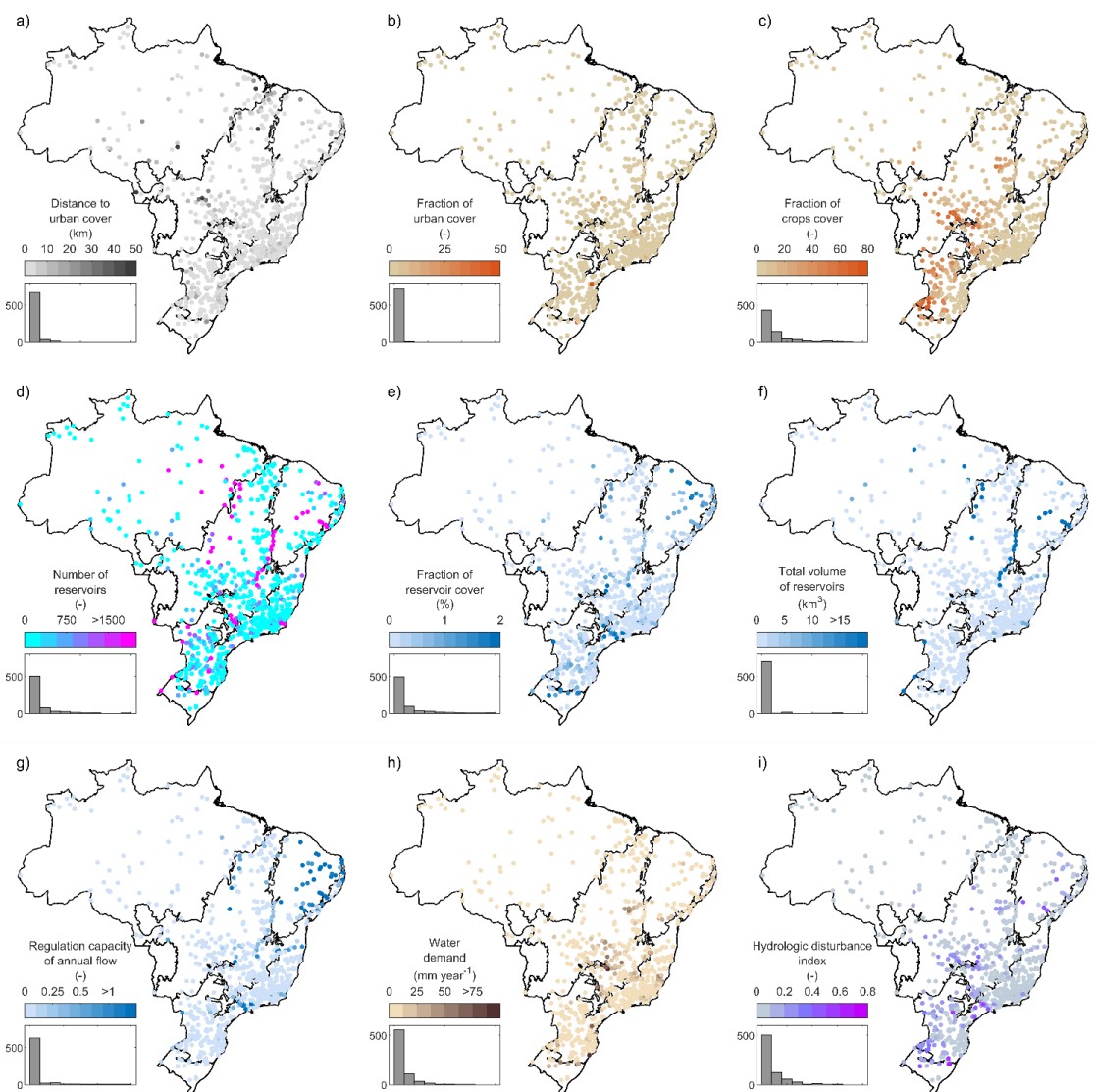

**Figure 13: Spatial distribution of the hydrologic disturbance attributes of CABra catchments. a. Distance from urban cover to the streamflow gauge, in km; b. Urban fraction of land-cover, in percentage; c. Crops fraction of land-cover, in percentage; d. The number of reservoirs in the catchment; e. Reservoir fraction of land-cover, in percentage; f. The total volume of the reservoirs in the catchment, in km³; g. The capacity of the reservoirs in the catchment to regulate the mean annual streamflow, dimensionless; h. Multi-purpose water demand in the catchment, in mm year⁻¹; i. Hydrologic disturbance index (HDI) of the catchment, dimensionless. The HDI is a weighted relationship between all the anthropogenic factors of the catchments.**

The random forest regressor algorithm (Figure 14) showed us the most relevant hydrological signatures captured by the Hydrologic Disturbance Index. About 25% of the variance of the HDI is explained by the Half-flow day and the Streamflow Elasticity, which are two signatures extremely sensitive to streamflow regulation and to the generation of runoff in the catchment. Our results show us that the index is capable to capture what it was intended to: catchments with higher values presents a large number or high regulation capacity of reservoirs, or a great percentage of non-natural areas. Medium values

present some level of non-natural areas (pasture or crops), but there is not a high hydrological disturbance. Finally, lower values of HDI indicates minimally human-impacted catchments.

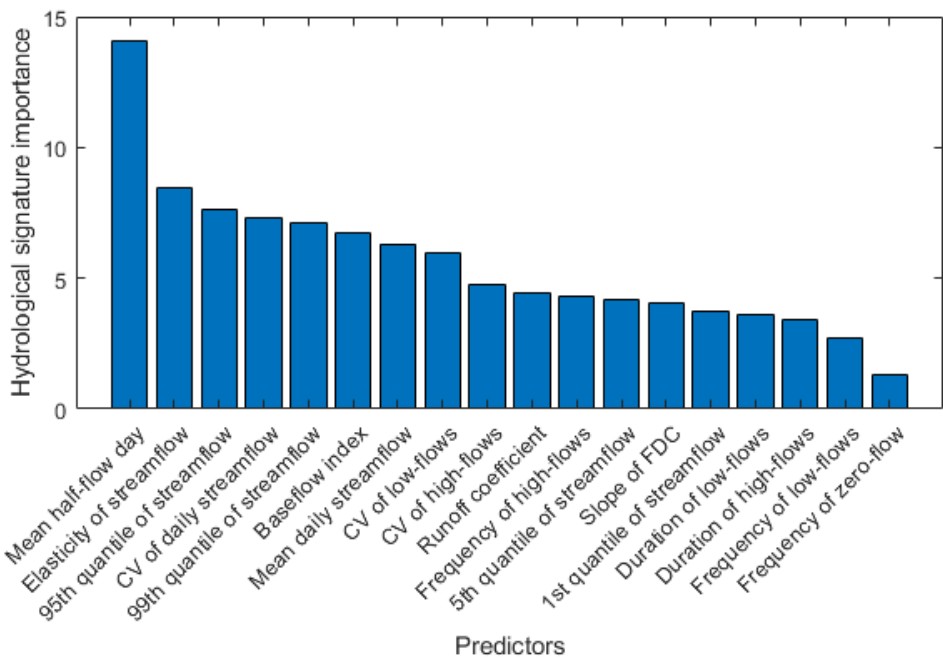


**Figure 14: Hydrological signatures as predictors of the Hydrological Disturbance Index. The random forest regressor algorithm assess how much each signature increase the error of a HDI prediction when randomly sorted. The higher the deviaton caused by a predictor, the higher is the influence of the hydrological signature on the HDI.**

### 2.9.3 Uncertainty and limitations

Uncertainties in hydrological disturbance are mainly related to the components of the index. As mentioned before, there is a limitation of use in the land-cover maps for small villages, urban areas, fragmented areas, and transitional areas of croplands, due to the spatial resolution of the land-cover maps. Because of this, small areas of urban fraction ($U_C$), and consequently the distance to the urban area ($U_D$), and crops area ($CR_C$), might be undetected and this fraction of the index – representing 40% – disconsidered or underestimated. Another 50% of the HDI is derived from reservoir data, from the ANA database.

Although the reservoirs data have been extensively improved through the years, there are still uncertainties related to the many sources of them. Different sources does not use the same satellite products or methodology to identify and catalog the reservoirs. Additionally, latest inclusions of reservoirs were automatically made and there were not a quality check of these data. Due to the crucial importance of reservoirs to the HDI, unrealistic number, areas and volumes of reservoirs can lead to unrealistic values of the index. The last component considered here is the water demand ($W_D$), is a area-averaged estimation,

which accounts to both consumptive and non-consumptive water abstractions, possible leading to higher values than real abstractions. Even representing, 10% of HDI composition, it should be taken in account in post-processing.

## 3 Comparison with the CAMELS-BR and broader implications for hydrological studies

The CABra and the CAMELS-BR (Chagas et al., 2020) contain both large samples of hydroclimatic, landscape, and other attributes for Brazilian catchments. Their striking similarities in concept and goals highlight nothing but the urgent need for the creation of such a database for Brazilian catchments. However, it is important to notice that multiple differences between both datasets exist, as we will discuss below.

The first main difference between CABra and CAMELS-BR is related to the catchment delineation procedures adopted. CAMELS-BR uses the basin masks from the GSIM (Do et al., 2018) product, where a 500-m digital elevation model was used for the delineation of catchment boundaries and extraction of topographic indices. GSIM has a quality filter allowing for up to 50% of error in the catchment area when compared with ANA's value, as described in Do et al. (2018). As previously explained, the CABra catchment boundaries (delineated using streamflow gauge location from ANA), uses a high-definition (90-m) elevation product. We have manually inspected each of the 735 catchments to minimize further errors, correcting the geographic position of the outlet to coincide with the stream network, achieving a mean error of 2% against ANA's areas. It is important to highlight that a suitable watershed delineation is of paramount importance for catchment hydrology studies because errors in these processes are further propagated for all computed attributes dependents on area and location. In addition, we provide the drainage network or CABra catchments.

Related to the daily streamflow data, in the CABra dataset we have retained catchments with less than 10% missing streamflow records over 30 hydrologic years (1980-2010) which resulted in the final selection of 735 catchments. On the other hand, CAMELS-BR contains 897 catchments with less than 5% missing data, while considering 20 hydrologic years, (1990-2009). Additionally, CAMELS-BR also provide longer timeseries when available for the gauge. Our choice for a longer time series was predicated on the commonly adopted rationale which assumes 30 years as the basis for establishing long-term climatology as well as hydrologic indices (Huntingford et al., 2014; Tetzlaff et al., 2017), which we in turn believe will lead to better characterization of hydrological and climatological processes taking place. A correlation test between hydrological signatures of 607 overlapping catchments in CABra and CAMELS-BR datasets is shown in Figure 15. The signatures based only in daily streamflow values, such as daily mean streamflow (q_mean), $5^{th}$ and $95^{th}$ quantiles of daily streamflow (q_5 and q_95), are quite similar between CABra and CAMELS-BR, showing that both periods of analysis were capable to capture the streamflow patterns of the catchments. When comparing signatures related to frequency and duration of low and high streamflow events, we can note little variation but still good agreement between datasets. In this case, the distinct period for hydrological signatures calculation (1980-2010 in CABra, and 1990-2009 in CAMELS-BR) might be the cause of deviations. The slope of flow duration curve and runoff coefficient are in a very good agreement ($r^2 > 0.95$), demonstrating that both datasets are using precipitation products with good reliability. The streamflow elasticity and baseflow index have presented notable differences between CABra and CAMELS-BR. This might be due to the different components adopted in the equations of Woods (2009) and Ladson et al. (2013), which were implemented for elasticity and baseflow index calculations.

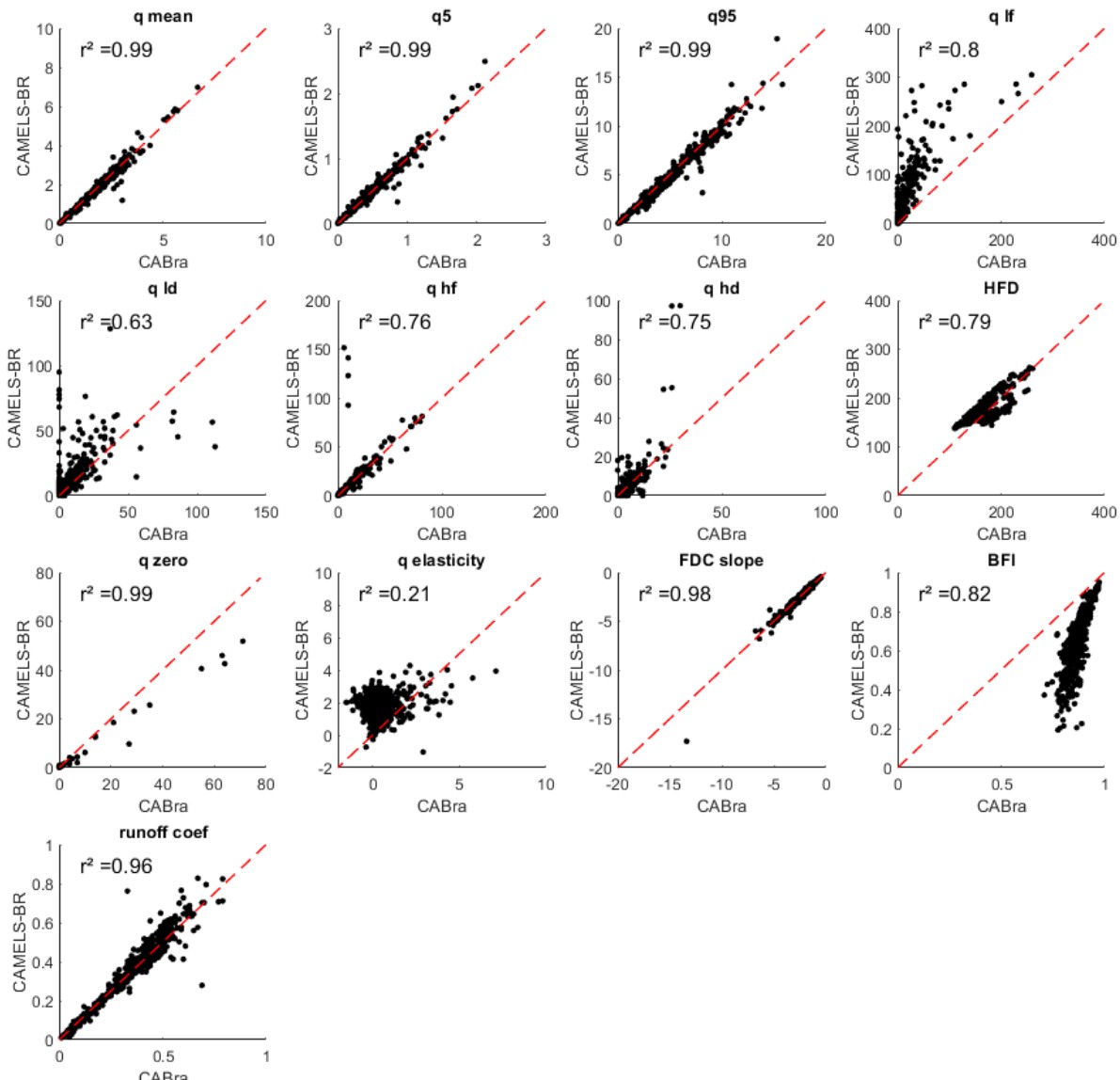

785

**Figure 15: Scatter plots and correlation coefficients between hydrological signatires of CABra and CAMELS-BR catchments. There was 607 catchmnets and 13 hydrological signatures overlapped in both datasets.**

Another important difference between both datasets is related to the choice of databases used for providing the daily meteorological time series and estimated the related indices. While CAMELS-BR uses three widely used gridded datasets (based on remote sensing/reanalysis/gauge blends of rainfall), i.e., the CHIRPS v2.0, CPC, and MSWEP v2.2, being the first one the chosen for the climatic indices (because of its spatial resolution of 0.05°x0.05°), the CABra uses the Xavier et al. (2016) dataset and the ERA5 reanalysis. The Xavier et al. (2016) dataset was produced based on observations from 3,625

790

rain gauges and 735 wheatear stations in the Brazilian territory and is extensively used as the ground-truth reference for the validation of precipitation products, including the CHIRPS, MSWEP, and the soil moisture satellite-corrected estimates (SM2RAIN, Brocca et al. (2014)) (Paredes-Trejo et al., 2018), the Global Precipitation Measurement (GPM, Hou et al. (2014)) (Gadelha et al., 2019), the Tropical Rainfall Measuring Mission (TRMM, Huffman et al. (2007)) (Melo et al., 2015). Other uses of this dataset include the evaluation of precipitation from downscaled-global circulation models (Almagro et al., 2020), as well as other meteorological variables used in regional studies (Battisti et al., 2019; Bender and Sentelhas, 2018; Monteiro et al., 2018), aside from being widely used for hydrological studies (Almagro et al., 2017; Avila-Diaz et al., 2020; Lima and AghaKouchak, 2017; Souza et al., 2016). The main limitation of Xavier's dataset it that it covers only Brazil.

Additional differences belonging to the meteorological time-series section are also worth noting. CAMELS-BR provides the model-based PET estimates extracted from the GLEAM product (Martens et al., 2017), while daily temperatures (maximum, minimum, and average) are the only PET-related variable provided in a daily time series format. The CABra dataset provides the computed PET following 3 widely used methods, along with all necessary variables for its computation, such as solar radiation, wind speed, temperature, and relative humidity. Our choice for the computation of PET instead of using model-based estimates should allow for more transparency and reproducibility of results obtained using our dataset. Also, the choice of providing a wider range of meteorological variables allows the user to estimate PET based on different methods while enhancing the reach of our dataset for studies that might benefit from additional meteorological variables.

While the soil and geology attributes of both CABra and CAMELS-BR are derived from the same data sources, (i.e., the SoilGrids250, the GLiM, and the GLHYMPS v2.0), CABra provides the following additional variables not available in CAMELS-BR: saturated permeability (saturated hydraulic conductivity for geology attribute), soil type, textural class, and soil bulk density – which can be used to estimate soil porosity. Regarding groundwater attributes, CABra contains rock type and name of the aquifer and water table depths from Fan et al., (2013) and the HAND estimates, while CAMELS-BR contains only the water table depth estimates from Fan et al., (2013).

In terms of land-cover attributes, CABra and CAMELS-BR present similar attributes, but the data source is different. CABra adopted a product with a higher spatial resolution (100-m against 300-m) and more recent observation (2015 against 2009) than in CAMELS-BR. Due to this better spatial resolution. we chose to use a most recent land cover, even it being outside of the timespan of hydrologic time series. CABra also brings information about the seasonal vegetation biomass of the catchment, in terms of NDVI, which is not present in CAMELS-BR.

Finally, both datasets take into account the human influence within each catchment, which is essential to a holistic understanding of the catchment behavior due to anthropogenic interactions and a lack of most of the large-sample datasets (Addor et al., 2020). CAMELS-BR presents data about water use, the volume of reservoirs, and the degree of regulation of the reservoirs. However, there is no combination or integration of these attributes in a specific index or approach. On the other hand, CABra presents eight attributes, i.e., distance to urbanization, the fraction of non-natural land-cover (crops and urban areas), water demand, reservoirs' count, area, volume, and streamflow regulation capacity (the last two are also found in CAMELS-BR), which can affect the hydrologic behavior of the catchment in terms of water quantity, quality and

regulation. Additionally, we developed a new hydrologic disturbance index (HDI), which considers all these eight attributes above-mentioned. The HDI is a quantitative index of the level of anthropization, being reproducible and practical to identify a more or less human-impacted catchment.

## 4 Conclusions

In this study, we have collected, synthesized, organized, and made available more than 100 topography, climate, streamflow, groundwater, soil, geology, land-use, and land cover, and hydrologic disturbance attributes for 735 catchments in Brazil. To do so, we have used several sources, such as observed time series, observed and modeled gridded data, remote sensing data, and reanalysis data. Moreover, we have calculated some attributes for providing more accurate data than those available in the literature, including potential evapotranspiration, and providing inexistent data, such as the hydrological disturbance index. As this dataset deals with catchment-scale averaged attributes, we have paid particular attention to DEM resolution, catchment delineation, while also manually inspecting each of the CABra catchments.

The development of the CABra dataset opens up several opportunities to test and develop a hypothesis in a unique environment like Brazil, with its vast and rich diversity in hydrology and landscapes. Finding relationships between the catchments' attributes will enable hydrologists to identify the drivers of the water fluxes in the catchment. We hope our dataset will aid catchment classification efforts that will ultimately unravel the underlying dominant controls of Brazilian regional hydrology across space and time. At the same time, the CABra dataset covers fundamentally different hydroclimatologic and ecologic regions than those covered by other similar large-sample datasets (United States, Great Britain, Chile, etc.), being a complement for global assessments and expanding the possibility of the use of our dataset for multiple scientific areas, such as geology, agronomy, ecohydrology.

We intend to expand the CABra dataset in the future. Information and attributes related to relevant fields of work, such as soil erosion, ecology, biology, and chemistry, as well as climate change projections, will be added to the CABra dataset in future updates release. Thus, CABra represents a robust multi-source data collection effort for Brazil and is intended to play a key role in advancing the scientific understanding of climate-landscape-hydrology interactions. As such, we hope it will guide large-sample hydrology investigations and pave the way for testing novel hypotheses by both the Brazilian and the international scientific community.


**Data availability**

The datasets underlying the CABra dataset are available at https://doi.org/10.5281/zenodo.4070146. We also developed a website with a friendly interface for easy access by users: https://thecabradataset.shinyapps.io/CABra/.

**Author contribution**

AA, PTSO, AAMN, and PT conceived the ideas and designed the methodology for the study; AA collected, processed, and analyzed the data; AA, PTSO, and AAMN led the writing of the initial draft; TR and PT edited and reviewed the manuscript;
All authors contributed and gave final approval for publication.

**Competing interests**

The authors declare that they have no conflict of interest.

**Acknowledgments**

This study was supported by grants from the Ministry of Science, Technology, and Innovation – MCTI and National Council
for Scientific and Technological Development – CNPq [grants numbers 441289/2017-7, 306830/2017-5, and 309752/2020-5]. This study was also financed in part by the Coordenação de Aperfeiçoamento de Pessoal de Nível Superior - Brasil (CAPES) - Finance Code 001 and CAPES Print.

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
