# Peer review of "CABra: a novel large-sample dataset for Brazilian catchments"

_Hydrology and Earth System Sciences, 2020_

## Referee Comment (RC1) · Pedro Luiz Borges Chaffe (Referee) · 9 Dec 2020

This manuscript provides a dataset of catchment attributes for several Brazilian catchments. The dataset (including climate, streamflow, groundwater, and others) was compiled from several data sources and most of the methods and limitations are discussed in the specific sections. The data set is made public through Zenodo. The authors put a lot of effort in delineating the major basins, providing meteorological datasets, calculating potential evaporation with three different methods, and developing a new hydrologic disturbance index. I agree with the authors that "...similarities [with CAMELS-BR (Chagas et al., 2020)] highlight nothing but the urgent need for the creation of such a database for Brazilian catchments". Overall the manuscript is well written and it is worthy of publication after minor revisions.

[Figure]

As major comments, I believe the authors could highlight those new products they produced and the novelty of the manuscript by: (1) framing CABra as a complementary dataset that builds on some opportunities left by the CAMELS-BR dataset (e.g., need for drainage density and uncertainty analysis); (2) making use of the CAMELS-BR for a quantitative comparison to show if or how the different approaches and data sources might influence signature values. I would also create a new section after the Introduction called "2 Motivation to extend the CAMELS-BR data set" similarly to Addor et al. (2017). This new section would be a more concise and accurate (see minor comments) replacement of section "3 Comparison with the CAMELS-BR..." of the current manuscript.

Please, find bellow some minor comments:

L27-28 – References are highlighted in gray.

L35-39 – Why isn't CAMELS-BR cited here with all the other CAMELS datasets?

L46-47 – "Additionally, there is no repository...". How about CAMELS-BR?

L49-51 – This sentence might seem contradictory as CAMELS-BR (and perhaps CABra) had already been developed.

L67-72 – If this paragraph corresponds to Fig.1, should it read "hydrologic disturbance" or "Hydrologic signature"?

L109 – "100 cell accumulating water" is equivalent to what area?

L115 – Can you specify somewhere what are the 132 catchments?

L118-121 – This is not accurate. Do et al. (2018) and Gudmundsson et al. (2018) methodology was based on the areas provided in ANA's dataset (the same you used to check the error). Even though they probably did not visually inspect the boundaries for all the basins they provided, in the CAMELS-BR every boundary from the Do et al. (2018) data set was visually inspected (this procedure may not be explicit in Chagas et

al., 2020). Please, check those references.

L129-130 – What do you mean by "necessitating a better understanding of hydrologic processes"?

L165-172 – I think this ensemble can be another nice product of your data set if better described. Do you think that an ensemble mean product is always "more reliable"? Could you compare and show the differences?

L217 – "estimative" I think it should be "estimation".

L242-243 – "There are only a few. . . that have precipitation in the winter". I think you mean "most of the precipitation".

L244 – "Amazonian coastal area might not be obvious to the international readership. If this is really relevant, perhaps you could provide an indication in some figure.

L266-267 – What are those outliers? Are they relevant?

L280-282 – Many streamflow gauges have inconsistencies other than those typographical errors cited here. For example, there are abrupt changes resulting from changes in measurement instruments or rating curves, and unrealistic daily streamflow values (i.e., larger than 1000 mm d-1) (Chagas et al, 2020). Have you screened the time series for those inconsistencies? How do you think they would affect the hydrological signatures?

L298-299 – I might have missed in your paper how you defined a hydrologic year. Was it the same for the entire country?

L314-316 – I am not sure if "reaching infinity" is the best expression. I think it means we should not calculate the slope if the value in the denominator is zero.

L351-354 – It is nice that you provided HAND data. The "robust correlations" were found in this work or in Nobre et al.? If the latter, I think it is better to provide the reference again to avoid ambiguity.

L367-370 – You do not need this fractured rocks vs porous rocks discussion here.

L373-347 – You can also delete "This kind of. . . thereby forming rivers and lakes".

L415 – "There is a spatial correlation. . ." Why is it so? Is it an underlying process or a feature of how the data source was produced?

L417 – "we have" should be "there is".

L417-419 – "These characteristics. . . present the opposite". I would delete it or rephrase it.

L478 – You use satellite observations of 2015 but calculate the signatures using 1980-2010 data. It could be argued that this land-cover data does not correspond to the same period. Can you provide the rationale behind your choice?

L480-489 – You should highlight more this NDVI product that you provide as it is a nice addition.

L496 – Perhaps it would be better to use "forest and grasslands".

L508 – Is the MATOPIBA region really significant for discussion?

L509-511 – Is this a feature or a choice for the data set?

L521 – "Higher values were found in timing with. . ." – please rephrase it.

L539-540 – What is the data source for the reservoirs? The ANA (2017) reference was not provided in the reference list. In the CAMELS-BR the reservoir data from ANA (2018) and GRaND (Lehner et al., 2011) were combined and further checked against Pekel et al. (2016) in order to exclude very small (insignificant) reservoirs.

L542-543 – It is not clear the meaning and how you calculated "distance to the nearest urban area of each catchment". Distance to the outlet? How do you define urban area? An isolated urban pixel is considered an urban area or spurious data? How can this distance affect the streamflow signatures?

L545-551 – While the Hydrologic Disturbance index is an interesting and novel product, you do not explain how you determined those coefficients in Eq.6. You should investigate how to evaluate the usefulness of the index against the hydrologic signatures that you calculated. What does it capture? Can we use it to somehow classify what you observed?

Section 3 Comparison with the CAMELS-BR and broader implications for hydrological studies – In some parts of this section you provided advantages of CABra over CAMELS-BR, but you did not point out the limitations of CABra. Therefore, I do not consider it to be a "comparison" section. I would suggest you to make this section more concise and to create a new section "2 Motivation to extend the CAMELS-BR data set" similarly to Addor et al. (2017).

L602-603 – Even though you did a better job than CAMELS-BR at delineating basin boundaries, I do not think that a 2% error compared to the ANA values is the best or most correct standard for two reasons: (1) the methodology used by ANA to calculate the areas is not provided; (2) the areas provided by ANA are many times rounded to the nearest hundred or thousand.

L606-609 – "while considering 20 hydrologic years" – should read 20 or more hydrologic years. Even though the attributes were calculated using 1990-2009 (for consistency with other CAMELS data set), data for 1980-2010 was also provided when available. You should also clearly explain that some of the non typographical error types that were checked in CAMELS-BR (e.g., abrupt change, zero in place of missing data etc) might not have been checked in CABra.

L613-620 – Xavier et al. (2016) is a great interpolated product that has been used extensively. However, you should clarify the limitations of choosing to use Xavier et al. (2016) as you cited in the climate section of your paper. Since that data is interpolated inside Brazil, you cannot use this rainfall data sets in basins such as Amazon, Paraguay and Parana. Besides, Xavier et al. (2016) used many rainfall gauges (which is great)

without checking the homogeneity of the data. This is not a criticism at all. I just want to point out that there are advantages and disadvantages for each choice along the way and here lies the opportunity for a comparison of how different are the attributes calculated based on different data sources.

L625-632 – I think this is a great point of your paper (PET calculations) that should be better highlighted. Could other climate indices that were not provided in CABra (timing, frequency, duration, etc) be relevant for hydrological analysis?

L633 – "attributes of from" delete "from"

L633-638 – Could the second geological class or the relative percentages be relevant for hydrological analysis?

L639-643 – Does a higher spatial resolution always mean a smaller classification error?

L644-650 – This is a very nice addition that was not covered in the CAMELS-BR and I think you should highlight it more. You should also try to better describe and test this index. See comments to L539-540 (I do not think ANA provides reservoir sizes) and L545-551.

TABLE 4 – What is the meaning and how did you calculate q_hf (Max streamflow frequency) and q_lf (Min streamflow frequency)?

FIGURE 2 – What are the shades of blue in (b)? "Km" should be in lower case.

FIGURE 3 – (a) it is difficult to differentiate the order of smaller catchments. In this case, I think it is better to use a point at the outlet of each catchment. Some of the opportunities to expand the Chagas et al. (2020) data set were to provide drainage data and you could use that to highlight this important part of your work. It is not clear that the x labels of the inset are the same as the colorbars, perhaps you should increase the tick marks. You chose to keep the Biome delineation in the background, is there a specific reason for that choice? It might not be obvious to the international readership.

References Addor, N., Newman, A. J., Mizukami, N., and Clark, M. P.: The CAMELS data set: catchment attributes and meteorology for large-sample studies, Hydrol. Earth Syst. Sci., 21, 5293–5313, https://doi.org/10.5194/hess-21-5293-2017, 2017.

ANA – Brazilian National Water Agency: Relatorio de Seguranca de Barragens 2017, 2018. Chagas, V. B. P., Chaffe, P. L. B., Addor, N., Fan, F. M., Fleischmann, A. S., Paiva, R. C. D. and Siqueira, V. A.: CAMELS-BR: hydrometeorological time series and landscape attributes for 897 catchments in Brazil, Earth Syst. Sci. Data, 12(3), 2075–2096, doi:10.5194/essd-12-2075-2020, 2020.

Do, H. X., Gudmundsson, L., Leonard, M., and Westra, S.: The Global Stream-flow Indices and Metadata Archive (GSIM) – Part 1: The production of a daily streamflow archive and metadata, Earth Syst. Sci. Data, 10, 765–785, https://doi.org/10.5194/essd-10-765-2018, 2018.

Gudmundsson, L., Do, H. X., Leonard, M., and Westra, S.: The Global Stream-flow Indices and Metadata Archive (GSIM) – Part 2: Quality control, time-series indices and homogeneity assessment, Earth Syst. Sci. Data, 10, 787–804, https://doi.org/10.5194/essd-10-787-2018, 2018.

Lehner, B., Liermann, C. R., Revenga, C., Vörösmarty, C., Fekete, B., Crouzet, P., Döll, P., Endejan, M., Frenken, K., Magome, J., Nilsson, C., Robertson, J. C., Rödel, R., Sindorf, N., and Wisser, D.: High-resolution mapping of the world's reservoirs and dams for sustainable river-flow management, Front. Ecol. Environ., 9, 494–502, https://doi.org/10.1890/100125, 2011.

Pekel, JF., Cottam, A., Gorelick, N. et al. High-resolution mapping of global surface water and its long-term changes. Nature 540, 418–422 (2016).
* * *

---

## Referee Comment (RC2) · Anonymous Referee #2 · 22 Jan 2021

This study has collected, synthesized, organized, and made available more than 100 topography, climate, streamflow, groundwater, soil, geology, land-use/land cover, and hydrological disturbance attributes for 735 catchments in Brazil. The dataset is valuable for many hydrological and other relevant scientific studies. However, this paper does not provide any in-depth/innovative scientific analysis based on the established dataset. It looks more like a dataset paper rather than a scientific research paper. I would like to recommend a rejection in HESS but would like to suggest a transfer to the journals focusing on data such as Earth System Science Data, Scientific Data - Nature, etc. More specific comments are as follows: 1) Some attributes are from first-hand investigation data (e.g., streamflow, meteorology, geology, catchment delineation, etc.), and a lot of attributes are extracted from global re-analysis dataset. The

former attributes owe higher accuracy and are really valuable for the community to do a variety of hydrometeorological modeling and assessment. The latter attributes are also useful to reduce other uses' time consumption to re-prepare them. But I would like to suggest providing the delineated catchment-based digital map and/or tabular dataset. The spatially distributed dataset is important for distributed modeling. 2) For groundwater attribute: spatial variation of groundwater table is subject to complicated driving factors. Fan et al.'s (2013) can be useful for a regional scale analysis for pattern recognition but would have a lot of uncertainties at catchment scale. The observed well observation dataset would be more useful for such a large-sample catchment dataset effort. 3) Typos to be checked: L44, than? L301 the all the year? L303 be showed seen?

---

## Author Comment (AC1) · 18 Feb 2021

Response to Anonymous Referee #2

Comment #1: This study has collected, synthesized, organized, and made available more than 100 topography, climate, streamflow, groundwater, soil, geology, land-use/land cover, and hydrological disturbance attributes for 735 catchments in Brazil. The dataset is valuable for many hydrological and other relevant scientific studies. However, this paper does not provide any in-depth/innovative scientific analysis based on the established dataset. It looks more like a dataset paper rather than a scientific research paper. I would like to recommend a rejection in HESS but would like to suggest a transfer to the journals focusing on data such as Earth System Science Data,

Scientific Data - Nature, etc.

Author's response #1: We would like to thank the referee for the insightful comments, suggestions, and kind words in support of our manuscript. The manuscript has been revised in accordance with your comments, which were highly insightful and enabled us to improve the quality of our manuscript. With the updated manuscript, we took care to answer the reviewer's questions, comments, and requests. We hope that the revisions in the manuscript and our accompanying responses will now meet the requirements for publication. Thank you again for your consideration.

Regarding to the transfer to another journal focusing on data, we disagree with referee. Our dataset is focused on hydrology, providing a large variety of catchment attributes that may contribute to the advancement of hydrological modelling, process concepts, besides the fact that all provided data is analysed in the study. The main inspiration for the CABra dataset is the MOPEX (Duan et al., 2006) and CAMELS-US datasets (Addor et al., 2017), which were published in journals focused in hydrology. The last one also collected, organized, synthetised, and analysed a wide range of catchment attributes for 671 catchments in continental US, being universally used and cited since its publication. We also have seen many other datasets being published in Hydrology and Earth System Sciences journal. Oubeidillah et al. (2014) made available a dataset of post-calibrated model parameters for hydroclimate impact assessment, where authors collected and organized data including meteorological forcings, soil, land-cover, vegetation and elevation from the best-available source at that time. Siebert et al. (2015) provided a global dataset of historical evolution of irrigated areas, collected from sub-national irrigations statistics from a variety of sources. For German territory, Zink et al. (2017) provided a high-resolution dataset of land surface variables. A new global land-based product of precipitation was made available by Contractor et al. (2020), by merging multiple archives of in situ data. The well-known ERA-Interim/Land reanalysis dataset (Balsamo et al., 2015) was also published in HESS journal, providing one of the biggest archives on land surface variables in the world. Finally, the CAMELS-CL

(Alvarez-Garreton et al., 2018), which is a catchment-based dataset of geophysical attributes, similar to CABra, for Chile was also published in HESS journal. Given the above we believe that CABra dataset is within the scope of HESS journal for publication, since we have not only organized existent data, but also presented novel products (e.g., hydrological disturbance index, gridded climate ensemble, and potential evapotranspiration) and conducted deep analysis of the data.

Comment #2: More specific comments are as follows: 1) Some attributes are from first-hand investigation data (e.g., streamflow, meteorology, geology, catchment delineation, etc.), and a lot of attributes are extracted from global re-analysis dataset. The former attributes owe higher accuracy and are really valuable for the community to do a variety of hydrometeorological modeling and assessment. The latter attributes are also useful to reduce other uses' time consumption to re-prepare them. But I would like to suggest providing the delineated catchment-based digital map and/or tabular dataset. The spatially distributed dataset is important for distributed modeling.

Author's response #2: Thank you for the comment. We agree with referee that the spatially distributed information about the attributes provided by CABra is of great importance for distributed modelling. But for such large dataset as CABra is quite unworkable to provide the catchment-based digital map for 735 catchments due to its large amount of data. To attend referee's requirement, we will add a table (attached to the dataset files in Zenodo and attached in this response) indicating the link to download each of the digital maps used for CABra's development. Since we already provide the catchment boundary, it is easy to the user to download and clip the digital map using a GIS application.

Comment #3: 2) For groundwater attribute: spatial variation of groundwater table is subject to complicated driving factors. Fan et al.'s (2013) can be useful for a regional scale analysis for pattern recognition but would have a lot of uncertainties at catchment scale. The observed well observation dataset would be more useful for such a large-sample catchment dataset effort.

Author's response #3: Thank you for your comment and suggestion. We will add to the dataset the static and dynamic levels observed in 2010, which is the last year from the daily timeseries of climate and streamflow in CABra dataset (Figure R1). We have obtained the wells observations at the Geological Survey of Brazil – CPRM database for groundwater, the Groundwater Information System – SIAGAS (http://siagasweb.cprm.gov.br/).

Comment #4: 3) Typos to be checked: L44, than? L301 the all the year? L303 be showed seen?

Author's response #4: Thanks for noting! We removed "than" at L44 (L45 of track change revised manuscript). We corrected "through the all the year" to "through the year" at L301 (L338 of track change revised manuscript). We removed "showed" at L303 (L340 of track change revised manuscript).

REFERENCES

Addor, N., Newman, A. J., Mizukami, N. and Clark, M. P.: The CAMELS data set: catchment attributes and meteorology for large-sample studies, Hydrol. Earth Syst. Sci., 21, 5293–5313, doi:10.5194/hess-21-5293-2017, 2017.

Alvarez-Garreton, C., Mendoza, P. A., Pablo Boisier, J., Addor, N., Galleguillos, M., Zambrano-Bigiarini, M., Lara, A., Puelma, C., Cortes, G., Garreaud, R., McPhee, J. and Ayala, A.: The CAMELS-CL dataset: Catchment attributes and meteorology for large sample studies-Chile dataset, Hydrol. Earth Syst. Sci., 22(11), 5817–5846, doi:10.5194/hess-22-5817-2018, 2018.

Balsamo, G., Albergel, C., Beljaars, A., Boussetta, S., Brun, E., Cloke, H., Dee, D., Dutra, E., Munõz-Sabater, J., Pappenberger, F., De Rosnay, P., Stockdale, T. and Vitart, F.: ERA-Interim/Land: A global land surface reanalysis data set, Hydrol. Earth Syst. Sci., 19(1), 389–407, doi:10.5194/hess-19-389-2015, 2015.

Contractor, S., Donat, M. G., Alexander, L. V., Ziese, M., Meyer-Christoffer, A., Schneider, U., Rustemeier, E., Becker, A., Durre, I. and Vose, R. S.: Rainfall Estimates on a Gridded Network (REGEN) - A global land-based gridded dataset of daily precipitation from 1950 to 2016, Hydrol. Earth Syst. Sci., 24(2), 919–943, doi:10.5194/hess-24-919-2020, 2020.

Duan, Q., Schaake, J., Andréassian, V., Franks, S., Goteti, G., Gupta, H. V., Gusev, Y. M., Habets, F., Hall, a., Hay, L., Hogue, T., Huang, M., Leavesley, G., Liang, X., Nasonova, O. N., Noilhan, J., Oudin, L., Sorooshian, S., Wagener, T. and Wood, E. F.: Model Parameter Estimation Experiment (MOPEX): An overview of science strategy and major results from the second and third workshops, J. Hydrol., 320(1–2), 3–17, doi:10.1016/j.jhydrol.2005.07.031, 2006.

Oubeidillah, A. A., Kao, S. C., Ashfaq, M., Naz, B. S. and Tootle, G.: A large-scale, high-resolution hydrological model parameter data set for climate change impact assessment for the conterminous US, Hydrol. Earth Syst. Sci., 18(1), 67–84, doi:10.5194/hess-18-67-2014, 2014.

Siebert, S., Kummu, M., Porkka, M., Döll, P., Ramankutty, N. and Scanlon, B. R.: A global data set of the extent of irrigated land from 1900 to 2005, Hydrol. Earth Syst. Sci., 19(3), 1521–1545, doi:10.5194/hess-19-1521-2015, 2015.

Zink, M., Kumar, R., Cuntz, M. and Samaniego, L.: A high-resolution dataset of water fluxes and states for Germany accounting for parametric uncertainty, Hydrol. Earth Syst. Sci., 21(3), 1769–1790, doi:10.5194/hess-21-1769-2017, 2017.

Please also note the supplement to this comment: https://hess.copernicus.org/preprints/hess-2020-521/hess-2020-521-AC1-supplement.pdf

[Figure]

**Fig. 1.** Figure R1

---

## Author Comment (AC2) · 18 Feb 2021

Response to Referee #1 - Pedro Luiz Borges Chaffe

Comment #1: This manuscript provides a dataset of catchment attributes for several Brazilian catchments. The dataset (including climate, streamflow, groundwater, and others) was compiled from several data sources and most of the methods and limitations are discussed in the specific sections. The data set is made public through Zenodo. The authors put a lot of effort in delineating the major basins, providing meteorological datasets, calculating potential evaporation with three different methods, and developing a new hydrologic disturbance index. I agree with the authors that ". . .similarities [with CAMELS-BR (Chagas et al., 2020)] highlight nothing but the urgent need

for the creation of such a database for Brazilian catchments". Overall the manuscript is well written and it is worthy of publication after minor revisions.

Author's response #1: We would like to thank the referee for the insightful comments, suggestions, and kind words in support of our manuscript. The manuscript has been revised in accordance with your comments, which were highly insightful and enabled us to improve the quality of our manuscript. With the updated manuscript, we took care to answer the reviewer's questions, comments, and requests. We hope that the revisions in the manuscript and our accompanying responses will now meet the requirements for publication. Thank you again for your consideration.

Comment #2: As major comments, I believe the authors could highlight those new products they produced and the novelty of the manuscript by: (1) framing CABra as a complementary dataset that builds on some opportunities left by the CAMELS-BR dataset (e.g., need for drainage density and uncertainty analysis); (2) making use of the CAMELS-BR for a quantitative comparison to show if or how the different approaches and data sources might influence signature values. I would also create a new section after the Introduction called "2 Motivation to extend the CAMELS-BR data set" similarly to Addor et al. (2017). This new section would be a more concise and accurate (see minor comments) replacement of section "3 Comparison with the CAMELS-BR. . ." of the current manuscript.

Author's response #2: We appreciate your comments. (1) This dataset design and construction started a way before the publication of the CAMELS-BR dataset as seen in the first public presentation of the CABra dataset, in Oliveira et al. (2020). So, it was not "built on opportunities left by the CAMELS-BR dataset". As most of catchment attributes datasets worldwide, CABra and CAMELS-BR present some overlapping attributes, especially the primary geophysical information of the catchments. Even so, most of these attributes were obtained from different sources. We agree that both datasets are complementary, but CABra was not designed to fill gaps on CAMELS-BR but inspired by CAMELS-US and MOPEX. (2) Thanks for suggesting this analysis. We

have performed (see Fig. R1) a simple but effective analysis to verify in which hydrological signatures and how the different approaches may impact more. We performed a correlation test in 607 corresponding catchments in both datasets. As can be seen in the figure, the signatures based only in daily streamflow values, such as daily mean streamflow (q_mean), 5th and 95th quantiles of daily streamflow (q5 and q95), are quite similar between CABra and CAMELS-BR, showing that both periods of analysis were capable to capture the streamflow patterns of the catchments. When comparing signatures related to frequency and duration of low and high streamflow events, we can note little variation but still good agreement between datasets. In this case, the distinct period for hydrological signatures calculation (1980-2010 in CABra, and 1990-2009 in CAMELS-BR) might be the cause of deviations. The slope of flow duration curve and runoff coefficient are in a very good agreement ($r^2 > 0.95$), demonstrating that both datasets are using precipitation products with good reliability. The streamflow elasticity and baseflow index have presented notable differences between CABra and CAMELS-BR. This might be due to the different components adopted in the equations of Woods (2009) and Ladson et al. (2013), which were implemented for elasticity and baseflow index calculations. This discussion can be found at L687-701 of the track change revised manuscript. (3) We are thankful for the suggestion, but we disagree with referee's comment. The section "2 Motivation to extend the Newman et al. (2015) data set" of Addor et al. (2017) is in a completely different context. The CAMELS-US dataset is really an extension of the Newman et al. (2015) dataset, which provides daily meteorological forcing (from three different sources) and daily streamflow (from USGS) for 671 catchments in US. From these 671 catchments previously defined by Newman et al. (2015), a wide range of geophysical attributes were made available Addor et al. (2017). To derive the climatic indices, authors also used the Newman et al. (2015) daily meteorological timeseries. To derive the hydrological signatures, authors also used the Newman et al. (2015) daily streamflow timeseries. Summing up, the dataset provided in Newman et al. (2015) was taken as a basis for the development of CAMELS-US, and more than that, it is part of CAMELS-US. In turn, the

CAMELS-BR were not used for any process/derivation in or as basement for the CABra dataset development, which is not an extension of any other dataset. We only used the CAMELS-BR to a comparison of the attributes and its sources, presented in "3 Comparison with the CAMELS-BR dataset". This is like the section "9 Comparison with the MOPEX data set" found in Addor et al. (2017), where authors compare their newly developed dataset with a previously developed dataset. There, authors compare the period and source of observations, criteria to include or not a catchment, similarities, dissimilarities, advantages, and disadvantages of each dataset. This is the same approach we have made in the section mentioned by the referee. Given the above, we think that the most appropriate is to keep the section "3 Comparison with the CAMELS-BR dataset", only making the corrections and improvements suggested throughout this revision.

Please, find bellow some minor comments: Comment #3: L27-28 – References are highlighted in gray.

Author's response #3: Thanks for noting! We corrected this.

Comment #4: L35-39 – Why isn't CAMELS-BR cited here with all the other CAMELS datasets?

Author's response #4: As CAMELS-BR were cited in a specific paragraph for Brazilian catchments datasets in L49-54, we preferred to not cite the CAMELS-BR in the previous paragraphs. But for address the referee's demand, we will add the CAMELS-BR in the list of worldwide CAMELS datasets (L39 of track change revised manuscript).

Comment #5: L46-47 – "Additionally, there is no repository. . .". How about CAMELS-BR?

Author's response #5: Thanks for the comment! Some parts of our manuscript were written before the publication of the CAMELS-BR dataset (Chagas et al., 2020). Despite this, the abovementioned statement should have been corrected for the submission. We apologize the error and we corrected it in the revised version of the CABra manuscript (L47-48 of track change revised manuscript).

Comment #6: L49-51 – This sentence might seem contradictory as CAMELS-BR (and perhaps CABra) had already been developed.

Author's response #6: This sentence was written there to state that CABra dataset was being developed before the CAMELS-BR (discussion and published versions). Despite this, we agree with referee's comment and we have modified the sentence to: "Recently, two large-sample datasets for catchment attributes were developed for Brazil: ..." (L50 of track change revised manuscript).

Comment #7: L67-72 – If this paragraph corresponds to Fig.1, should it read "hydrologic disturbance" or "Hydrologic signature"?

Author's response #7: "Hydrologic disturbance" is the correct, once we are describing the attribute classes of CABra dataset, but we agree that the information is not well-presented in the actual form of the paragraph. For the revised version, we have modified the figure to better explain the study design (L74 of track change revised manuscript).

Comment #8: L109 – "100 cell accumulating water" is equivalent to what area?

Author's response #8: Considering that the MERIT DEM has $\sim$90m (at Equator), a 100-cell accumulation is $\sim$0.81 km$^2$.

Comment #9: L115 – Can you specify somewhere what are the 132 catchments?

Author's response #9: Sure. The location map of the 132 mentioned catchments is attached in Figure R2.

Comment #10: L118-121 – This is not accurate. Do et al. (2018) and Gudmundsson et al. (2018) methodology was based on the areas provided in ANA's dataset (the same you used to check the error). Even though they probably did not visually inspect the

boundaries for all the basins they provided, in the CAMELS-BR every boundary from the Do et al. (2018) data set was visually inspected (this procedure may not be explicit in Chagas et al., 2020). Please, check those references.

Author's response #10: Thanks for the information! We agree that Do et al. (2018) probably was not visually inspected due to the large number of basins worldwide. We only state that CAMELS-BR were not manually inspected because in Chagas et al. (2020) authors says "The main limitation of the procedure of Do et al. (2018) is that catchment boundaries were not manually inspected" and any information indicating that this limitation has been improved is presented to the reader, suggesting that the limitation persists in CAMELS-BR. Anyway, we corrected the information by removing the CAMELS-BR of the citation of non-checked datasets (L116-121 of track change revised manuscript).

Comment #11: L129-130 – What do you mean by "necessitating a better understanding of hydrologic processes"?

Author's response #11: A need for a better understanding of the hydrologic processes on different scales (local to continental). After referee's comment we realized that the sentence is not well-presented to the reader and we will modify it to "necessitating a better understanding of hydrologic processes on different scales" (L131 of track change revised manuscript).

Comment #12: L165-172 – I think this ensemble can be another nice product of your data set if better described. Do you think that an ensemble mean product is always "more reliable"? Could you compare and show the differences?

Author's response #12: Thanks for the comment and suggestion. We have improved the presentation of the ensemble product in the revised version of the manuscript (see L171-175 of track change revised manuscript). We believe that it is not appropriate to say that an ensemble is always better than individual products, but it is a fact that an ensemble will avoid unrealistic values generated by individual products and are an

important tool to overcome spatial lack of stations in ground-based products, such as Xavier et al. (2016) dataset used in CABra. Newman et al. (2015b) found that ensemble product of precipitation and temperature still capture the main features of the variables and, moreover, improves the identification of extreme event frequency. When talking about climate projections, it is know that an ensemble generally outperforms individual forecasts (Bellucci et al., 2015; Solman et al., 2013; Tebaldi et al., 2005), being capable to detect internal variability and seasonal patterns. In CABra dataset, the main goal of creating such ensemble was to create a product covering all the catchments extension, merging two high-resolution and high-quality products that englobes all desired climate variables. The plot of the catchments in the Budyko space can be seen in the attached Figure R3, for REF, ERA5, and ENS climate datasets, respectively. We can note that the main climate features are captured by all the datasets, with catchments in Caatinga being more arid, followed by the Cerrado. The Atlantic Forest is in the same location at the Budyko space, while some catchments in Amazon only appears on ERA5 and ENS dataset, due to its extension outside REF.

Comment #13: L217 – "estimative" I think it should be "estimation".

Author's response #13: Thanks for noting. We have changed in the main text (L226 of track change revised manuscript).

Comment #14: L242-243 – "There are only a few. . . that have precipitation in the winter". I think you mean "most of the precipitation".

Author's response #14: Actually, our statement is correct. Most of the catchments in Brazil presents the precipitation in timing with the temperature (L240-242) and we highlight the small group of catchments in Brazil which precipitation is not in timing with the temperature in L242-243, as can be seen in Figure 4b.

Comment #15: L244 – "Amazonian coastal area might not be obvious to the international readership. If this is really relevant, perhaps you could provide an indication in some figure.

Author's response #15: Thanks for the comment. We agree with reviewer's note. For most of the international readers, the Amazon is restricted to an inland forest. Despite this, the coastal area of the Amazon is of main importance for the large amounts of precipitation in the forest, with more than 300 days per year with sea-breeze induced precipitation.

Comment #16: L266-267 – What are those outliers? Are they relevant?

Author's response #16: The outliers are catchments in which the long-term water balance cannot be explained only by the Budyko hypothesis. In these group of catchments there must be some feature that is not taken in account in the Budyko hypothesis that is controlling the long-term water balance, making these catchments truly relevant for hydrologic studies due to its unknow features outside the Budyko context (evaporation, potential evaporation, and precipitation). From these catchments, CABra users are able to conduct further investigations, leading to advance in the hydrologic processes understanding.

Comment #17: L280-282 – Many streamflow gauges have inconsistencies other than those typographical errors cited here. For example, there are abrupt changes resulting from changes in measurement instruments or rating curves, and unrealistic daily streamflow values (i.e., larger than 1000 mm d-1) (Chagas et al, 2020). Have you screened the time series for those inconsistencies? How do you think they would affect the hydrological signatures?

Author's response #17: We also noted those inconsistences on streamflow gauges data. Some of them were addressed, but not all of them. We believe that these information is missing in our manuscript and we added them on the revised version (L302-309 of track change revised manuscript). Inconsistences as referee commented can completely modify the hydrologic signatures, e.g., daily streamflow larger than 1000 mm.d-1 will lead to an overestimation of signatures based on mean values (mean daily flow, aridity index, runoff ratio). When these values are repeated for a long time, they
can modify signatures based on the frequency and of streamflow, e.g., flow duration curve, high and low flows frequency and duration. We checked for outliers on the streamflow data by comparing each value 'x' to its neighbours. Elements with value larger than five times the median of a sliding ten-elements window (centred in 'x') were considered as an invalid value (NaN).

Comment #18: L298-299 – I might have missed in your paper how you defined a hydrologic year. Was it the same for the entire country?

Author's response #18: We defined the hydrologic year as the October 1st – September 30th period, as cited in L307. Considering referee's comment, we presented this information before mentioning any result considering the hydrologic year (L314-316 of track change revised manuscript). Although it is know that in Brazil we have different precipitation cycles periods (Almagro et al., 2020), we considered the same hydrologic year for the whole country, as adopted by the Brazilian Water Agency in their annual reports (ANA, 2020a). Moreover, many other hydrological studies in Brazil have been used the same hydrological period for its analysis (Alvalá et al., 2019; Cunha et al., 2019; de Jesus et al., 2020; Lucas et al., 2015; Marcuzzo and Goularte, 2013; Marengo et al., 2013; Neto et al., 2016).

Comment #19: L314-316 – I am not sure if "reaching infinity" is the best expression. I think it means we should not calculate the slope if the value in the denominator is zero.

Author's response #19: Thanks for the comment. We agree with the referee. In the revised version we have changed it to: "In our analyses, we also found zero values between the 33rd and 66th percentiles of the slope of flow duration curve in the northeastern portion of Brazil, in the Caatinga biome, which indicates the existence of catchments with non-perennial rivers in that region, which are mainly dependent on direct runoff of rainfall." (L351-353 of track change revised manuscript).

Comment #20: L351-354 – It is nice that you provided HAND data. The "robust correlations" were found in this work or in Nobre et al.? If the latter, I think it is better to

provide the reference again to avoid ambiguity.

Author's response #20: We appreciate the comment. In the abovementioned discussion, the "robust correlations" were found in our study, especially when looking the spatial distribution. So, it is not a statistic correlation, but a visual one, showing similar distribution between HAND and Water Table Depth data.

Comment #21: L367-370 – You do not need this fractured rocks vs porous rocks discussion here.

Author's response #21: Thanks for the suggestion. We will remove the discussion.

Comment #22: L373-347 – You can also delete "This kind of. . . thereby forming rivers and lakes".

Author's response #22: Thanks for the suggestion. We will remove it from the discussion.

Comment #23: L415 – "There is a spatial correlation. . ." Why is it so? Is it an underlying process or a feature of how the data source was produced?

Author's response #23: We believe that this is an underlying process especially related to the soil characteristics. As can be seen in Figure 9 of the manuscript, there is a spatial correlation between organic carbon, depth to the bedrock and bulk density. Generally shallower and less compacted soils (high values of bulk density) present higher concentrations of carbon (Sena, 2016; Vezzani and Mielniczuk, 2011), especially in Ferrasols, Acrisols and Lixisols, which were found in that region.

Comment #24: L417 – "we have" should be "there is".

Author's response #24: Thanks for noting. We have changed in the main text (see L458 of track change revised manuscript).

Comment #25: L417-419 – "These characteristics. . . present the opposite". I would delete it or rephrase it.

Author's response #25: Thanks for the suggestion. We have changed it to: "These characteristics, allied to the favourable climate, turned this region attractive to agriculture." in L459-460 of track change revised manuscript.

Comment #26: L478 – You use satellite observations of 2015 but calculate the signatures using 1980- 2010 data. It could be argued that this land-cover data does not correspond to the same period. Can you provide the rationale behind your choice?

Author's response #26: We have chosen to use the 2015 observation from the PROBA-V satellite due to its high spatial resolution against other available products. In fact, we know that the period mismatch can lead to some uncertainty. We also know that the ideal is to provide a year-to-year land-cover map/attributes and that is a goal for futures updates of the CABra dataset, using a high-resolution product for Brazil, the MapBiomas (Souza et al., 2020), which was recently updated to the South-America extension.

Comment #27: L480-489 – You should highlight more this NDVI product that you provide as it is a nice addition.

Author's response #27: Thanks for the suggestion. We have improved the presentation of the NDVI products adopted in CABra dataset in the revised version of the manuscript (L522-524 of track change revised manuscript).

Comment #28: L496 – Perhaps it would be better to use "forest and grasslands".

Author's response #28: Thanks for the suggestion. We changed "grass" to "grassland" (see L539 of track change revised manuscript).

Comment #29: L508 – Is the MATOPIBA region really significant for discussion?

Author's response #29: Thanks for the comment. MATOPIBA is the larger agricultural frontier in Brazil, but we agree that it is not relevant for the discussion, especially for an international reader. We removed the MATOPIBA from the discussion.

Comment #30: L509-511 – Is this a feature or a choice for the data set?

Author's response #30: It is a feature from the dataset, based on the land-cover attributes.

Comment #31: L521 – "Higher values were found in timing with. . ." – please rephrase it.

Author's response #31: We rephrased it to "Higher values of NDVI occurs in the accordance to the seasonal cycle, . . ." in L564-565 of track change revised manuscript.

Comment #32: L539-540 – What is the data source for the reservoirs? The ANA (2017) reference was not provided in the reference list. In the CAMELS-BR the reservoir data from ANA (2018) and GRaND (Lehner et al., 2011) were combined and further checked against Pekel et al. (2016) in order to exclude very small (insignificant) reservoirs.

Author's response #32: Thanks for noting. It is true that the reference for the reservoirs' source is missing. We will provide it on the revised version. To clarify, in our database, we have used the National Water Mass Reference Database v2019, available at: https://metadados.snirh.gov.br/geonetwork/srv/eng/catalog.search#/metadata/7d054e5a-8cc9-403c-9f1a-085fd933610c. This database was prepared for generating information to support actions for planning, managing, and regulating water resources in Brazil, as described in ANA (2020b).

Comment #33: L542-543 – It is not clear the meaning and how you calculated "distance to the nearest urban area of each catchment". Distance to the outlet? How do you define urban area? An isolated urban pixel is considered an urban area or spurious data? How can this distance affect the streamflow signatures?

Author's response #33: Thanks for the comment. We agree that it is not clear, and we improved this statement in the revised manuscript (L604 of track change revised manuscript). Yes, it is the distance from the catchment outlet to the nearest pixel of urban area derived from the land-cover map. The closer to an urban area, the more is

the chance that a streamflow is impacted by human activities. It is not a rule but can give us a preliminary view of the hydrological disturbance of the catchment.

Comment #34: L545-551 – While the Hydrologic Disturbance index is an interesting and novel product, you do not explain how you determined those coefficients in Eq.6. You should investigate how to evaluate the usefulness of the index against the hydrologic signatures that you calculated. What does it capture? Can we use it to somehow classify what you observed?

Author's response #34: Thanks for the comment and suggestion. Our goal was to create a simple index, with easily accessible inputs, that is capable to measure how much disturbed a catchment is in relation to its hydrology. Since the beginning of CABra development, it was known that most of the catchments were minimally urbanised, but with some of them with changes in the original land-cover (conversion of natural vegetation to cropland/pasture). Some studies conducted in Brazil found that, besides the fact of the interference by the conversion of natural vegetation to pasture, this led to minimal changes in the surface hydrology of the catchment, being more relevant to groundwater recharge and soil chemistry (Bacellar, 2005; Lanza, 2015; Nepstad et al., 1994; Salemi et al., 2012). Additionally, it has been seen that the human-induced impact of the reservoirs can be more relevant than the natural ones, and can significantly alter natural hydrological processes (Zhao et al., 2016), leading to an increase/decrease of streamflow and hydrological droughts characteristics (Wanders and Wada, 2015; Ye et al., 2003; Zhang et al., 2015). Moreover, Zhang et al. (2015) found that hydrologic vulnerability is also directly related to human water abstractions, but this can be compensated by streamflow regulation of the reservoirs. This led us to an integrated analysis of the reservoir regulation and human water abstract to reach the optimal balance on our index. Based on the abovementioned, we have decided to use weighted information about the land-cover, reservoirs, and water demand of each catchment. We considered the reservoir-based information with more impact: regulation capacity with 40%, number of reservoirs and its percentage of catchment area with

5% each. The second most impacting factor of the index is the non-natural land-cover in the catchment, which can lead to modify hydrological surface and subsurface processes, with 40% of the weights. Finally, the water abstraction of the catchment was pondered with 10%. We applied a random forest algorithm for a regression analysis (see the attached Figure R4) that showed us the most relevant hydrological signatures captured by the Hydrologic Disturbance Index. About 25% of the variance of the HDI is explained by the Half-flow day and the Streamflow Elasticity, which are two signatures extremely sensitive to streamflow regulation. Our results show us that the index is capable to capture what it was intended to: catchments with higher values presents a large number or high regulation capacity of reservoirs, or a great percentage of non-natural areas. Medium values present some level of non-natural areas (pasture or crops), but there is not a high hydrological disturbance. Finally, lower values of HDI indicates minimally human-impacted catchments.

Comment #35: Section 3 Comparison with the CAMELS-BR and broader implications for hydrological studies – In some parts of this section you provided advantages of CABra over CAMELS-BR, but you did not point out the limitations of CABra. Therefore, I do not consider it to be a "comparison" section. I would suggest you to make this section more concise and to create a new section "2 Motivation to extend the CAMELS-BR data set" similarly to Addor et al. (2017).

Author's response #35: Thanks for the comment. We made clear the major limitations of the CABra dataset in the revised manuscript. Regarding to the creation of a section "2 Motivation to extend the CAMELS-BR data set" similarly to Addor et al. (2017), as discussed before on this revision, we decide to keep the section "3 Comparison with the CAMELS-BR dataset", only making the corrections and improvements suggested throughout this revision. This is because the CABra dataset is not an extension of the CAMELS-BR. CABra is a newly developed large-sample dataset for Brazilian catchments, which did not used CAMELS-BR data for its development.

Comment #36: L602-603 – Even though you did a better job than CAMELS-BR at

delineating basin boundaries, I do not think that a 2% error compared to the ANA values is the best or most correct standard for two reasons: (1) the methodology used by ANA to calculate the areas is not provided; (2) the areas provided by ANA are many times rounded to the nearest hundred or thousand.

Author's response #36: Thanks for the comment. This approach could not be the best or most correct approach to state that the area calculation is closer or not to the real one, but this is the only official information we have in Brazil about the catchment areas. We agree that the ANA's values are rounded and the delineation methodology is not provided, but considering this lack of information, we have no other option than consider ANA's areas as the "real", since ANA is the only official database and the major maintainer/provider of the streamflow records in Brazil.

Comment #37: L606-609 – "while considering 20 hydrologic years" – should read 20 or more hydrologic years. Even though the attributes were calculated using 1990-2009 (for consistency with other CAMELS data set), data for 1980-2010 was also provided when available. You should also clearly explain that some of the non typographical error types that were checked in CAMELS-BR (e.g., abrupt change, zero in place of missing data etc) might not have been checked in CABra.

Author's response #37: Thanks for the comment. We know that the CAMELS-BR provides raw streamflow timeseries for a longer period than 1990-2009 (when available for the gauge), since it is clearly stated in Chagas et al. (2020). Despite this, in this section, we are only comparing the criteria adopted to retain only catchments with more consistent data, and therefore we chose to not include this information. To attend the referee's comment, we have changed the statement to "Related to the daily streamflow data, in the CABra dataset we have retained catchments with less than 10% missing streamflow records over 30 hydrologic years (1980-2010) which resulted in the final selection of 735 catchments. On the other hand, CAMELS-BR contains 897 catchments with less than 5% missing data, while considering 20 hydrologic years, (1990-2009). Additionally, CAMELS-BR also provide longer timeseries when available for the gauge"

in L681-684 of track change revised manuscript.

As mentioned on previous answers, we also have conducted a check on some non-typographical errors, but this was not clearly informed to the reader. This information and the methods used were included in the revised version of the manuscript as mentioned in Response #17.

Comment #38: L613-620 – Xavier et al. (2016) is a great interpolated product that has been used extensively. However, you should clarify the limitations of choosing to use Xavier et al. (2016) as you cited in the climate section of your paper. Since that data is interpolated inside Brazil, you cannot use this rainfall data sets in basins such as Amazon, Paraguay and Parana. Besides, Xavier et al. (2016) used many rainfall gauges (which is great) without checking the homogeneity of the data. This is not a criticism at all. I just want to point out that there are advantages and disadvantages for each choice along the way and here lies the opportunity for a comparison of how different are the attributes calculated based on different data sources.

Author's response #38: Thanks for the insightful comment. We totally agree that there are advantages and disadvantages in using the Xavier et al. (2015) and any other precipitation product. It is true that we can not use the Xavier's dataset in catchments that fall outside its extension (e.g., Amazon, Parana and Pantanal basins), and this is the major motivation to use a global dataset as the ERA5. Using the ERA5 dataset, we overcame the Xavier's limitation. Moreover, to blend the high quality of Xavier's dataset and the higher range of ERA5, we have created an ensemble mean covering all the CABra catchments. This way, we were able to use the Xavier's dataset within Brazilian territory, ERA5 dataset outside Brazilian territory, and an optimized precipitation dataset in all catchments, resulting in three daily climate files for each catchment. This allows the final users to decide the best product for your approach.

Comment #39: L625-632 – I think this is a great point of your paper (PET calculations) that should be better highlighted. Could other climate indices that were not provided in

CABra (timing, frequency, duration, etc) be relevant for hydrological analysis?

Author's response #39: Thanks for the comment. We worked on the improvement of the PET products presentation (see L190-192 of track change revised manuscript). In general, climatic indices are extremely relevant in hydrological analysis. Seo et al. (2019) related that there are specific subsets of climate indices that can be related to specific hydrological extremes, such as flood-events. Authors also state that the indices must be carefully selected and tested regarding the climate regime and the geophysical attributes of the area. Additionally, Renard and Thyer (2019) concluded that there are hidden climate indices that drive temporal and spatial variability of extreme occurrences in the catchments. Although CABra does not present all the widely used climatic indices (such those 27 proposed by the ETCCDI), the dataset is covered by the most influent climate indices, as shown in Addor et al. (2018). There, authors' results shows that aridity and seasonality and timing of precipitation (fraction of precipitation falling as snow may have no occurrence in Brazilian catchments) are the best predictors of hydrological signatures, especially mean annual discharge and half-flow date. Finally, even not containing all the mentioned climate indices, CABra climate attributes are capable to be potential predictors in hydrological analysis.

Comment #40: L633 – "attributes of from" delete "from"

Author's response #40: Thanks for noting. We have corrected it (L724 of track change revised manuscript).

Comment #41: L633-638 – Could the second geological class or the relative percentages be relevant for hydrological analysis?

Author's response #41: Not directly. Categorical attributes like the most and the second most common geological classes give us preliminary information and basis to some inference about the underlying processes of water movement on the surface and subsurface, e.g., sedimentary rocks generally present more porosity than metamorphic rocks. But this information will only be valid with the observed values of porosity, which

in CABra, are independent from the lithological class, varying spatially.

Comment #42: L639-643 – Does a higher spatial resolution always mean a smaller classification error?

Author's response #42: Many studies in the literature report that the classification accuracy is improved with finer spatial-resolution products, as found in Chen et al. (2004), Cushnie (1987), Fisher et al. (2018), and Huang et al. (2003). Moreover, Cushnie (1987) and Chen et al. (2004) states that the more heterogeneous are the land-cover, the finer resolution are required for a better accuracy and higher-resolution images generates lower spatial distribution classification errors. In CABra dataset, we have chosen to use the finest freely available land-cover product due to the high variability type of classes throughout Brazilian territory.

Comment #43: L644-650 – This is a very nice addition that was not covered in the CAMELS-BR and I think you should highlight it more. You should also try to better describe and test this index. See comments to L539-540 (I do not think ANA provides reservoir sizes) and L545-551.

Author's response #43: Thanks for the comment! This index came up from the need to an easy way to verify how much a catchment still contains of its original hydrological conditions and behaviour. The more the index, the more hydrological disturbed is the catchment. And we would like to create this using wide-available input parameters, so we chose to use land-cover, reservoirs, and water demand information, which are made available the most of national water agencies in the world. The reservoir sizes are provided in the shapefile related to the National Water Mass Reference Database v2019, available at: https://metadados.snirh.gov.br/geonetwork/srv/eng/catalog.search#/metadata/7d054e5a-8cc9-403c-9f1a-085fd933610c.

In the revised version of the manuscript, we have improved the presentation of the Hydrologic Disturbance Index in the main text (see L581-602 of track change revised

manuscript), using the information and analysis provided in the "Author's answer for Referee's comment on L545-551".

Comment #44: TABLE 4 – What is the meaning and how did you calculate q_hf (Max streamflow frequency) and q_lf (Min streamflow frequency)?

Author's response #44: The q_hf is the frequency of high flow events. The q_lf is the frequency of low flow events. They are calculated by computing the average number of days per year with flows equalling or not exceeding 20% the median of daily streamflow (for q_lf) and equalling or exceeding nine times the median of the daily streamflow (for q_hf). To clear up any doubt about the meaning of the hydrological signatures we have changed the description of the following attributes (Table 4) (L317-320 of track change revised manuscript). We also have inserted the description and methodology of all hydrological signatures presented in CABra dataset (L322-333 of track change revised manuscript).

Comment #45: FIGURE 2 – What are the shades of blue in (b)? "Km" should be in lower case.

Author's response #45: Thanks for noting, we changed "Km" to "km". There is no value related to the shade of blue. They are only the CABra catchments. As some catchments are within other ones, the coloured polygons are superimposed in the figure. The shades of blue were created to allow visualization of all catchment boundaries.

Comment #46: FIGURE 3 – (a) it is difficult to differentiate the order of smaller catchments. In this case, I think it is better to use a point at the outlet of each catchment. Some of the opportunities to expand the Chagas et al. (2020) data set were to provide drainage data and you could use that to highlight this important part of your work. It is not clear that the x labels of the inset are the same as the colorbars, perhaps you should increase the tick marks. You chose to keep the Biome delineation in the background, is there a specific reason for that choice? It might not be obvious to the international readership.

Author's response #46: Thanks for the suggestions. We have inserted a point to each catchment outlet as required by the referee. We will also make the drainage data available, appending the shapefile to the existent dataset in Zenodo.

Related to the comment about the histograms x ticks, to make clear it has the same values of the colorbar, we will insert in the figure legend the following: "Histograms have the same value of colorbars."

The biomes boundaries were kept in the background because throughout the manuscript, we discussed some of the results using the biomes to identify spatial patterns. Since biomes are supposed to be large areas with similar and uniform hydroclimatic dynamics and characteristics (Brown and Maurer, 1989; Coutinho, 2016) we expect that the similarities could be seen at the geophysical and hydroclimatic attributes of CABra.

REFERENCES

Addor, N., Newman, A. J., Mizukami, N. and Clark, M. P.: The CAMELS data set: catchment attributes and meteorology for large-sample studies, Hydrol. Earth Syst. Sci., 21, 5293–5313, doi:10.5194/hess-21-5293-2017, 2017.

Addor, N., Nearing, G., Prieto, C., Newman, A. J., Le Vine, N. and Clark, M. P.: A Ranking of Hydrological Signatures Based on Their Predictability in Space, Water Resour. Res., 54(11), 8792–8812, doi:10.1029/2018WR022606, 2018.

Almagro, A., Oliveira, P. T. S., Rosolem, R. and Hagemann, S.: Performance evaluation of Eta/HadGEM2-ES and Eta/MIROC5 precipitation simulations over Brazil, Atmos. Res., 244(1 November 2020), 105053, 2020.

Alvalá, R. C. D. S., Cunha, A. P. M. A., Brito, S. S. B., Seluchi, M. E., Marengo, J. A., Moraes, O. L. L. and Carvalho, M. A.: Drought monitoring in the Brazilian semiarid region, An. Acad. Bras. Cienc., 91, 1–15, doi:10.1590/0001-3765201720170209, 2019.

ANA: Conjuntura dos recursos hídricos no Brasil 2020: informe anual, Brasília. [online] Available from: http://conjuntura.ana.gov.br/static/media/conjuntura-completo.23309814.pdf, 2020a. ANA: Technical Note N. 52/2020/SPR, Brasília., 2020b.

Bacellar, L. de A. P.: O papel das florestas no regime hidrológico de bacias hidrográficas, Geo.br, 1, 1–39, 2005.

Bellucci, A., Haarsma, R., Gualdi, S., Athanasiadis, P. J., Caian, M., Cassou, C., Fernandez, E., Germe, A., Jungclaus, J., Kröger, J., Matei, D., Müller, W., Pohlmann, H., Salas y Melia, D., Sanchez, E., Smith, D., Terray, L., Wyser, K. and Yang, S.: An assessment of a multi-model ensemble of decadal climate predictions, Clim. Dyn., 44(9–10), 2787–2806, doi:10.1007/s00382-014-2164-y, 2015.

Brown, J. H. and Maurer, B. A.: Macroecology : The Division of Food and Space Among Species on Continents, Science (80-. )., 243, 1989.

Chagas, V. B. P., Chaffe, P. L. B., Addor, N., Fan, F. M., Fleischmann, A. S., Paiva, R. C. D. and Siqueira, V. A.: CAMELS-BR: hydrometeorological time series and landscape attributes for 897 catchments in Brazil, Earth Syst. Sci. Data, 12(3), 2075–2096, doi:10.5194/essd-12-2075-2020, 2020.

Chen, D., Stow, D. A. and Gong, P.: Examining the effect of spatial resolution and texture window size on classification accuracy: An urban environment case, Int. J. Remote Sens., 25(11), 2177–2192, doi:10.1080/01431160310001618464, 2004.

Coutinho, L. M.: Biomas brasileiros, 1st ed., Oficina de textos, São Paulo., 2016.

Cunha, A. P. M. A., Zeri, M., Leal, K. D., Costa, L., Cuartas, L. A., Marengo, J. A., Tomasella, J., Vieira, R. M., Barbosa, A. A., Cunningham, C., Cal Garcia, J. V., Broedel, E., Alvalá, R. and Ribeiro-Neto, G.: Extreme drought events over Brazil from 2011 to 2019, Atmosphere (Basel)., 10(11), doi:10.3390/atmos10110642, 2019.

Cushnie, J. L.: The interactive effect of spatial resolution and degree of internal variability within land-cover types on classification accuracies, Int. J. Remote Sens., 8(1), 15–29, doi:10.1080/01431168708948612, 1987.

Fisher, J. R. B., Acosta, E. A., Dennedy-Frank, P. J., Kroeger, T. and Boucher, T. M.: Impact of satellite imagery spatial resolution on land use classification accuracy and modeled water quality, Remote Sens. Ecol. Conserv., 4(2), 137–149, doi:10.1002/rse2.61, 2018.

Huang, H., Wu, B. and Fan, J.: Analysis to the Relationship of Classification Accuracy, Segmentation Scale, Image Resolution, Int. Geosci. Remote Sens. Symp., 6(C), 3671–3673, doi:10.1109/igarss.2003.1295233, 2003.

de Jesus, E. T., Amorim, J. da S., Junqueira, R., Viola, M. R. and de Mello, C. R.: Meteorological and hydrological drought from 1987 to 2017 in doce river basin, Southeastern Brazil, Rev. Bras. Recur. Hidricos, 25, 1–10, doi:10.1590/2318-0331.252020190181, 2020.

Ladson, A. R., Brown, R., Neal, B. and Nathan, R.: A standard approach to baseflow separation using the Lyne and Hollick filter, Aust. J. Water Resour., 17(1), 25–34, doi:10.7158/W12-028.2013.17.1, 2013.

Lanza, R.: Hidrologia comparativa e perda de solo e água em bacias hidrográficas cultivadas com eucalipto e campo nativo com pastagem manejada, Master Thesis, 150, 2015.

Lucas, M., Oliveira, P. T. S., Melo, D. C. D. and Wendland, E.: Evaluation of remotely sensed data for estimating recharge to an outcrop zone of the Guarani Aquifer System (South America), Hydrogeol. J., 23(5), 961–969, doi:10.1007/s10040-015-1246-1, 2015.

Marcuzzo, F. F. N. and Goularte, E. R. P.: Caracterização do Ano Hidrológico e Mapeamento Espacial das Chuvas nos Períodos Úmido e Seco do Estado do Tocantins, Rev. Bras. Geogr. Física, 06, 091–099, 2013.

Marengo, J. A., Alves, L. M., Soares, W. R., Rodriguez, D. A., Camargo, H., Riveros, M. P. and Pabló, A. D.: Two contrasting severe seasonal extremes in tropical South America in 2012: Flood in Amazonia and drought in Northeast Brazil, J. Clim., 26(22), 9137–9154, doi:10.1175/JCLI-D-12-00642.1, 2013.

Nepstad, D. C., Carvalho, C. R. De, Davidson, E. A., Jipp, P. H., Lefebvre, P. A., Negrelros, G. H., Sllva, E. D., Stone, T. A., Trumbore, S. E. and Vieira, S.: The role of deep roots in the hydrological and carbon cycles of Amazonian forests and pastures, Nature, 372(December), 666–669, 1994.

Neto, J. O. de A., Cota, G. E. M., Mendes, L. C., Magalhães, A. P. and Felippe, M. F.: Considerações sobre o ano hidrológico 2013- 2014 e os seus reflexos nos caudais fluviais da bacia do rio {Doce}, Rev. Geogr., 0(0), 26–45, 2016.

Newman, a. J., Clark, M. P., Sampson, K., Wood, a., Hay, L. E., Bock, a., Viger, R. J., Blodgett, D., Brekke, L., Arnold, J. R., Hopson, T. and Duan, Q.: Development of a large-sample watershed-scale hydrometeorological data set for the contiguous USA: Data set characteristics and assessment of regional variability in hydrologic model performance, Hydrol. Earth Syst. Sci., 19(1), 209–223, doi:10.5194/hess-19-209-2015, 2015a.

Newman, A. J., Clark, M. P., Craig, J., Nijssen, B., Wood, A., Gutmann, E., Mizukami, N., Brekke, L. and Arnold, J. R.: Gridded ensemble precipitation and temperature estimates for the contiguous United States, J. Hydrometeorol., 16(6), 2481–2500, doi:10.1175/JHM-D-15-0026.1, 2015b.

Oliveira, P. T. S., Almagro, A., Pitaluga, F., Meira Neto, A. A., Durcik, M. and Troch, P. A.: CABra: a novel large-scale dataset for Brazilian catchments, in AGU Fall Meeting, p. 12138., 2020.

Renard, B. and Thyer, M.: Revealing Hidden Climate Indices from the Occurrence of Hydrologic Extremes, Water Resour. Res., 55(9), 7662–7681,

doi:10.1029/2019WR024951, 2019.

Salemi, L. F., Groppo, J. D., Trevisan, R., Seghesi, G. B., Moraes, J. M., Ferraz, S. F. B. and Martinelli, L. A.: Consequências hidrológicas da mudança de uso da terra de floresta para pastagem na região da floresta tropical pluvial Atlântica, Ambient. e Agua - An Interdiscip. J. Appl. Sci., 7(3), 127–140, doi:10.4136/ambi-agua.927, 2012.

Sena, K. N.: COMPORTAMENTO DO CARBONO ORGÂNICO E DE ATRIBUTOS QUÍMICOS, FÍSICOS E MICROBIOLÓGICOS DE UM SOLO ARENOSO EM ÁREA DE CONVERSÃO PASTAGEM - EUCALIPTO, UNESP., 2016.

Seo, S. B., Kim, Y. O., Kim, Y. and Eum, H. Il: Selecting climate change scenarios for regional hydrologic impact studies based on climate extremes indices, Clim. Dyn., 52(3–4), 1595–1611, doi:10.1007/s00382-018-4210-7, 2019.

Solman, S. A., Sanchez, E., Samuelsson, P., da Rocha, R. P., Li, L., Marengo, J., Pessacg, N. L., Remedio, A. R. C., Chou, S. C., Berbery, H., Le Treut, H., de Castro, M. and Jacob, D.: Evaluation of an ensemble of regional climate model simulations over South America driven by the ERA-Interim reanalysis: Model performance and uncertainties, Clim. Dyn., 41(5–6), 1139–1157, doi:10.1007/s00382-013-1667-2, 2013.

Souza, C. M., Shimbo, J. Z., Rosa, M. R., Parente, L. L., Alencar, A. A., Rudorff, B. F. T., Hasenack, H., Matsumoto, M., Ferreira, L. G., Souza-Filho, P. W. M., de Oliveira, S. W., Rocha, W. F., Fonseca, A. V., Marques, C. B., Diniz, C. G., Costa, D., Monteiro, D., Rosa, E. R., Vélez-Martin, E., Weber, E. J., Lenti, F. E. B., Paternost, F. F., Pareyn, F. G. C., Siqueira, J. V., Viera, J. L., Neto, L. C. F., Saraiva, M. M., Sales, M. H., Salgado, M. P. G., Vasconcelos, R., Galano, S., Mesquita, V. V. and Azevedo, T.: Reconstructing three decades of land use and land cover changes in brazilian biomes with landsat archive and earth engine, Remote Sens., 12(17), doi:10.3390/RS12172735, 2020.

Tebaldi, C., Smith, R. L., Nychka, D. and Mearns, L. O.: Quantifying uncertainty in projections of regional climate change: A Bayesian approach to the analysis of multimodel

ensembles, J. Clim., 18(10), 1524–1540, doi:10.1175/JCLI3363.1, 2005.

Vezzani, F. M. and Mielniczuk, J.: Agregação e estoque de carbono em argissolo submetido a diferentes práticas de manejo agrícola, Rev. Bras. Cienc. do Solo, 35(1), 213–223, doi:10.1590/S0100-06832011000100020, 2011.

Wanders, N. and Wada, Y.: Human and climate impacts on the 21st century hydrological drought, J. Hydrol., 526, 208–220, doi:10.1016/j.jhydrol.2014.10.047, 2015.

Woods, R. A.: Analytical model of seasonal climate impacts on snow hydrology: Continuous snowpacks, Adv. Water Resour., 32(10), 1465–1481, doi:10.1016/j.advwatres.2009.06.011, 2009.

Xavier, A. C., King, C. W. and Scanlon, B. R.: Daily gridded meteorological variables in Brazil (1980-2013), Int. J. Climatol., 2659(October 2015), 2644–2659, doi:10.1002/joc.4518, 2015.

Ye, B., Yang, D. and Kane, D. L.: Changes in Lena River streamflow hydrology: Human impacts versus natural variations, Water Resour. Res., 39(7), 1–14, doi:10.1029/2003WR001991, 2003.

Zhang, R., Chen, X., Zhang, Z. and Shi, P.: Evolution of hydrological drought under the regulation of two reservoirs in the headwater basin of the Huaihe River, China, Stoch. Environ. Res. Risk Assess., 29(2), 487–499, doi:10.1007/s00477-014-0987-z, 2015.

Zhao, G., Gao, H., Naz, B. S., Kao, S. C. and Voisin, N.: Integrating a reservoir regulation scheme into a spatially distributed hydrological model, Adv. Water Resour., 98, 16–31, doi:10.1016/j.advwatres.2016.10.014, 2016.

[Figure]

**Fig. 1.** Figure R1. Scatter plots and correlation coefficients between hydrological signatires of CABra and CAMELS-BR catchments. There was 607 catchmnets and 13 hydrological signatures overlapped.

**Absolute error**

- **●** < 25%
- **●** 25% - 50%
- **●** 50% - 100%
- **●** 100% - 150%
- **●** 150% - 200%
- **●** > 200%
- ▨ Biomes

**Fig. 2.** Figure R2. Location of the 132 corrected outlets.

[Figure]

**Fig. 3.** Figure R3. Budyko space applied for data from REF, ERA5 and ENS climate datasets.

[Figure]

**Fig. 4.** Figure R4. Hydrological signatures importance for hydrological disturbance index by a random forest regressor algorithm.